# An acetate electrolyte for enhanced pseudocapacitve capacity in aqueous ammonium ion batteries

Zhuoheng Bao[1,3], Chengjie Lu[1,3], Qiang Liu[1], Fei Ye[1], Weihuan Li[1], Yang Zhou [1], Long Pan [1], Lunbo Duan[2], Hongjian Tang[2], Yuping Wu [2], Linfeng Hu [1]✉ & ZhengMing Sun [1]✉

Ammonium ion batteries are promising for energy storage with the merits of low cost, inherent security, environmental friendliness, and excellent electrochemical properties. Unfortunately, the lack of anode materials restricts their development. Herein, we utilized density functional theory calculations to explore the $V_2CT_x$ MXene as a promising anode with a low working potential. $V_2CT_x$ MXene demonstrates pseudocapacitive behavior for ammonium ion storage, delivering a high specific capacity of 115.9 mAh g$^{-1}$ at 1 A g$^{-1}$ and excellent capacity retention of 100% after 5000 cycles at 5 A g$^{-1}$. In-situ electrochemical quartz crystal microbalance measurement verifies a two-step electrochemical process of this unique pseudocapacitive storage behavior in the ammonium acetate electrolyte. Theoretical simulation reveals reversible electron transfer reactions with [NH$_4^+$(HAc)$_3$]⋯O coordination bonds, resulting in a superior ammonium ion storage capacity. The generality of this acetate ion enhancement effect is also confirmed in the MoS$_2$-based ammonium-ion battery system. These findings open a new door to realizing high capacity on ammonium ion storage through acetate ion enhancement, breaking the capacity limitations of both Faradaic and non-Faradaic energy storage.

The aqueous secondary batteries taking ammonium ions (NH$_4^+$) as charge carriers have captured tremendous attention in the sustainable energy storage research in recent years[1,2], which offer several notable advantages over traditional metallic carriers like Li$^+$, Na$^+$, K$^+$, Zn$^{2+}$, and Mg$^{2+}$: firstly, NH$_4^+$ exhibits fast diffusion ability in aqueous electrolytes owing to its small ionic size and light molar mass[3]; secondly, the utilization of NH$_4^+$ carriers ultimately eliminate the problems of dendritic growth which induces safety concerns in metal-ion batteries[4]; thirdly, NH$_4^+$ electrolytes are inexpensive and environmentally friendly. Taken together, the ammonium ion batteries (AIBs) are considered as promising candidates for practical, high-energy-density aqueous batteries. Recently, considerable efforts have been devoted to the development of cathode materials for ammonium ion storage. Ji et al. proposed a Prussian white analogue, namely, (NH$_4$)$_{1.47}$Ni[Fe(CN)$_6$]$_{0.88}$,

showing a specific capacity of 60 mAh g$^{-1}$ at 150 mA g$^{-1}$ (0.25 ‒ 1.5 V vs. standard hydrogen electrode, SHE) in (NH$_4$)$_2$SO$_4$ electrolyte[5]. Liu et al. developed the (NH$_4$)$_{0.27}$MnO$_{1.04}$(PO$_4$)$_{0.28}$ possessing a high specific capacity of 299.6 mAh g$^{-1}$ at 1 A g$^{-1}$ (0.2 ‒ 1.2 V vs. SHE) in ammonium acetate (NH$_4$Ac) electrolyte, with the coordination bonds between Mn atoms and acetate ions proposed to facilitate the ammonium ion storage process[6]. Our recent work first reported the ammonium ion storage behavior in some inorganic layered compounds including layered double hydroxides (183.7 mAh g$^{-1}$ at 0.1 A g$^{-1}$, 0 ‒ 1.2 V vs. SHE) and layered VOPO$_4$·2H$_2$O (154.5 mAh g$^{-1}$ at 0.1 A g$^{-1}$, −0.1 ‒ 1.2 V vs. SHE) with stable discharge plateau[7,8].

Although remarkable progress has been made on the cathode materials for AIBs most recently, its development for practical applications is still challenged by the lack of anode materials with a low

[1]School of Materials Science and Engineering, Southeast University, Nanjing 211189, P. R. China. [2]School of Energy and Environment, Southeast University, Nanjing 211189, P. R. China. [3]These authors contributed equally: Zhuoheng Bao, Chengjie Lu. ✉e-mail: linfenghu@seu.edu.cn; zmsun@seu.edu.cn

working voltage. Up to date, the development of anode materials is generally limited to transition metal oxide/sulfides (h-MoO$_3$[9], h-WO$_3$[10], etc.) and organic polymers (PTCDI[5], PANI[11], etc.). However, the low electronic conductivity of the former generally yields to unsatisfied specific capacity, and the high dissolution rate of the polymers results in poor cycling stability[12]. Undoubtedly, anode materials with a high capacity and excellent reversibility are in urgent need of development to construct full-cell AIBs for practical applications.

Since its first discovery and report[13], MXene, a new group of two-dimensional (2D) materials with excellent conductivity and high surface activity, has been proposed to be promising electrodes for energy storage[14–17]. Particularly, its broad 2D channel in the layered stacking is very desirable for ion storage compared with the narrow one-dimensional (1D) channel in transition metal oxide/sulfides which is incapable of accommodating large charge carriers[18,19]. Considering that V-based materials have relatively low working potential in AIBs[11,20,21], herein, we first carried out the theoretical simulation of V-based MXene by density functional theory (DFT) calculations using V$_2$CT$_x$ as an example[22,23]. The DFT calculation result verifies that V$_2$CT$_x$ MXene possesses the lowest working potential window compared with other V-based materials for AIBs reported, suggesting its potential as a very promising anode candidate for aqueous ammonium ion storage.

Inspired by this theoretical prediction, we tried to explore the ammonium ion storage behavior of V$_2$CT$_x$ MXene in various aqueous electrolytes including (NH$_4$)$_2$SO$_4$, NH$_4$Cl, (NH$_4$)$_2$C$_2$O$_4$, NH$_4$Me, NH$_4$Ac. Surprisingly, a pseudocapacitive typed redox reaction within −1 ~ −0.01 V potential range (vs. Ag/AgCl) could be only detected in the NH$_4$Ac electrolyte. Benefited from this pseudocapacitive behavior, our V$_2$CT$_x$ MXene delivered a high specific capacity of 115.9 mAh g$^{-1}$ (at 1 A g$^{-1}$), which surpasses all of the capacitive-typed electrodes in sustainable ammonium-ion batteries up to date. The pseudocapacitive origin was investigated by an in-situ electrochemical quartz crystal microbalance (EQCM) measurement, which reveals a two-step electrochemical process. Density functional theory (DFT) simulation further verifies the formation of [NH$_4$$^+$(HAc)$_3$]⋯O coordination bond facilitates the alternation of V$_2$CT$_x$ terminations, thereby realizing the variation of the V valence state which significantly promotes the charge transfer in the electrochemical process. The generality of this acetate ion enhancement effect on pseudocapacitive capacity was further confirmed in the MoS$_2$-based ammonium-ion battery system.

## Results

### Theoretical exploration of the V$_2$CT$_x$ MXene as an anode material

To identify the possibility of using V$_2$CT$_x$ MXene as a high-performance anode for AIBs, we have conducted a series of theoretical investigations based on DFT calculations. According to the previous report, the work function $\Phi$ which evaluates the energy required to activate an electron from the Fermi level to vacuum was proposed to be directly related to the electrochemical oxidation/reduction potential[24,25], thereby dominating the working potential. In the selection of electrodes, large work functional materials generally possess a wide working potential window to increase cell performance[26]. Therefore, we focus on the work function of a series of two-dimensional materials, including V$_2$C (trigonal), VO$_2$ (hexagonal)[27], VS$_2$ (hexagonal)[28], V$_2$O$_5$ (orthorhombic)[20], and V$_2$CT$_x$ MXene (Fig. 1a) and 1T-MoS$_2$. The plane-averaged electrostatic potential curves of V$_2$CT$_x$ (T = O, F, OH) are displayed in Fig. 1b−c and Supplementary Fig. 1. The values of work function were found to vary with surface terminations, determined to be 6.62 eV for V$_2$CO$_2$, 5.40 eV for V$_2$CF$_2$ and 1.85 eV for V$_2$C(OH)$_2$, respectively. Considering that the -OH termination was unlikely to adsorb the NH$_4$$^+$, the work function of V$_2$CT$_x$ (T contains O and F) was deduced to be in the range of 5.40 eV-6.62 eV. The work function of 1T-MoS$_2$ was determined to be 5.61 eV in Fig. 1d. As a consequence, the relationship between the work function and working potential window of investigated models is displayed in Fig. 1e, which indicates that the V$_2$CT$_x$ MXene possesses the largest work function and widest working potential window.

### Ammonium ion storage performance of V$_2$CT$_x$ MXene

On the basis of the above theoretical calculations, it is rational that V$_2$CT$_x$ is predicted to possess the lowest working potential window in

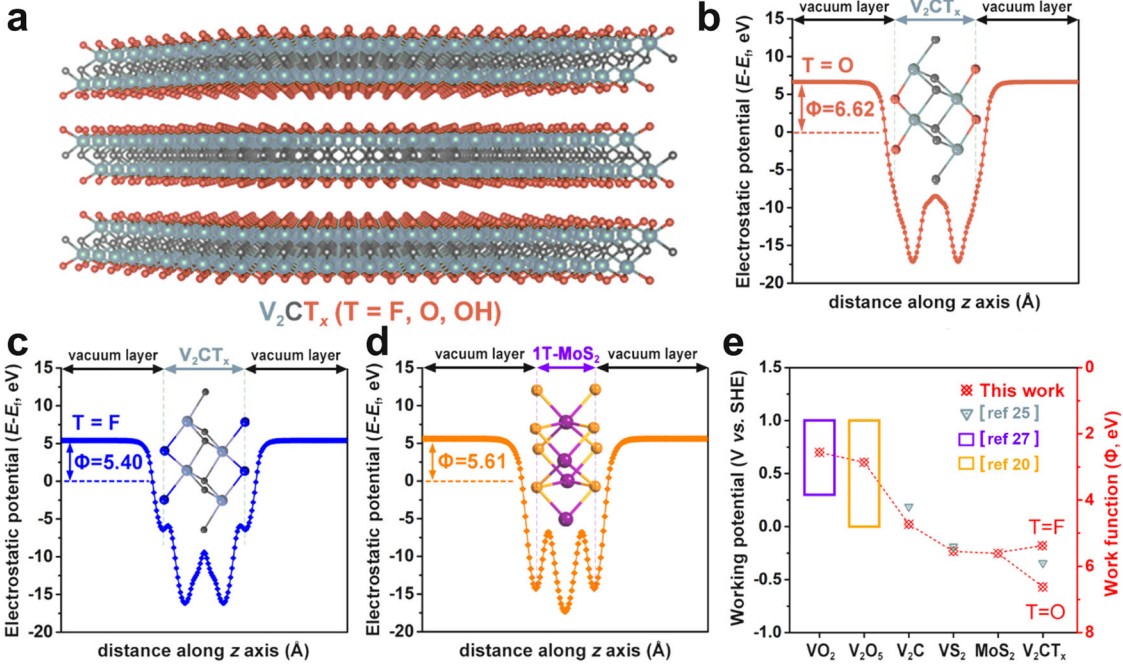

**Fig. 1 | DFT calculation. a** The side view of the structure model for V$_2$CT$_x$ MXene. Grey, black, and orange spheres denote V, C, and T terminations, respectively. **b, c** Plane-averaged electrostatic potential curve of V$_2$CO$_2$ and V$_2$CF$_2$, respectively. **d** Plane-averaged electrostatic potential curve of 1T-MoS$_2$. **e** Relationship between the work function and the working potential window of two-dimensional materials investigated in this work.

ammonium ion storage. Then, the $V_2AlC$ bulk precursor was prepared by pressureless sintering technique[28]. The multilayered-$V_2CT_x$ (m-$V_2CT_x$) was synthesized using the traditional LiF + HCl etching method[29]. Thereafter, the delaminated-$V_2CT_x$ (d-$V_2CT_x$) was obtained using tetraethyl ammonium hydroxide (TBAOH) delamination treatment. All products are characterized using X-ray Diffraction (XRD), and the collected patterns are displayed in Fig. 2a, showing a typical MXene structure of a layered hexagonal phase. Moreover, the diffraction peak located at the low-angle ($2\theta < 10°$) range originates from the diffraction of the (0002) crystal plane. Accordingly, a slight interlayer distance increase from 11.6 Å of m-$V_2CT_x$ to 12.0 Å of d-$V_2CT_x$ can be deduced after the TBAOH delamination treatment. The morphology of m-$V_2CT_x$ MXene was then observed using transmission electron microscopy (TEM) and scanning electron microscopy (SEM), showing a representative accordion-like block as

given in Supplementary Fig. 2. The d-$V_2CT_x$ was found to be semi-transparent under TEM observation in Fig. 2b, implying ultrathin structure with two-dimensional layer morphology. The surface area of the as-prepared MXene powders was measured by the Brunauer-Emmett-Teller (BET) method using nitrogen adsorption, which demonstrated a significant rise of specific surface area after the delamination process, from $0.996\,m^2\,g^{-1}$ for m-$V_2CT_x$ to as high as $26.002\,m^2\,g^{-1}$ for d-$V_2CT_x$ (Supplementary Fig. 3a). The high specific surface area is generally believed to favor the exposure active sites for ion adsorption or reaction. In addition, the micropores (size of which is smaller than 20 nm) obtained during the etching process played an important role in ion transportation through the sheet-like d-$V_2CT_x$ (Supplementary Fig. 3b). Clear lattice fringe in Fig. 2b was observed in the high-resolution transparent electron microscopy (HRTEM) image, and a spacing of 0.255 nm corresponded closely to

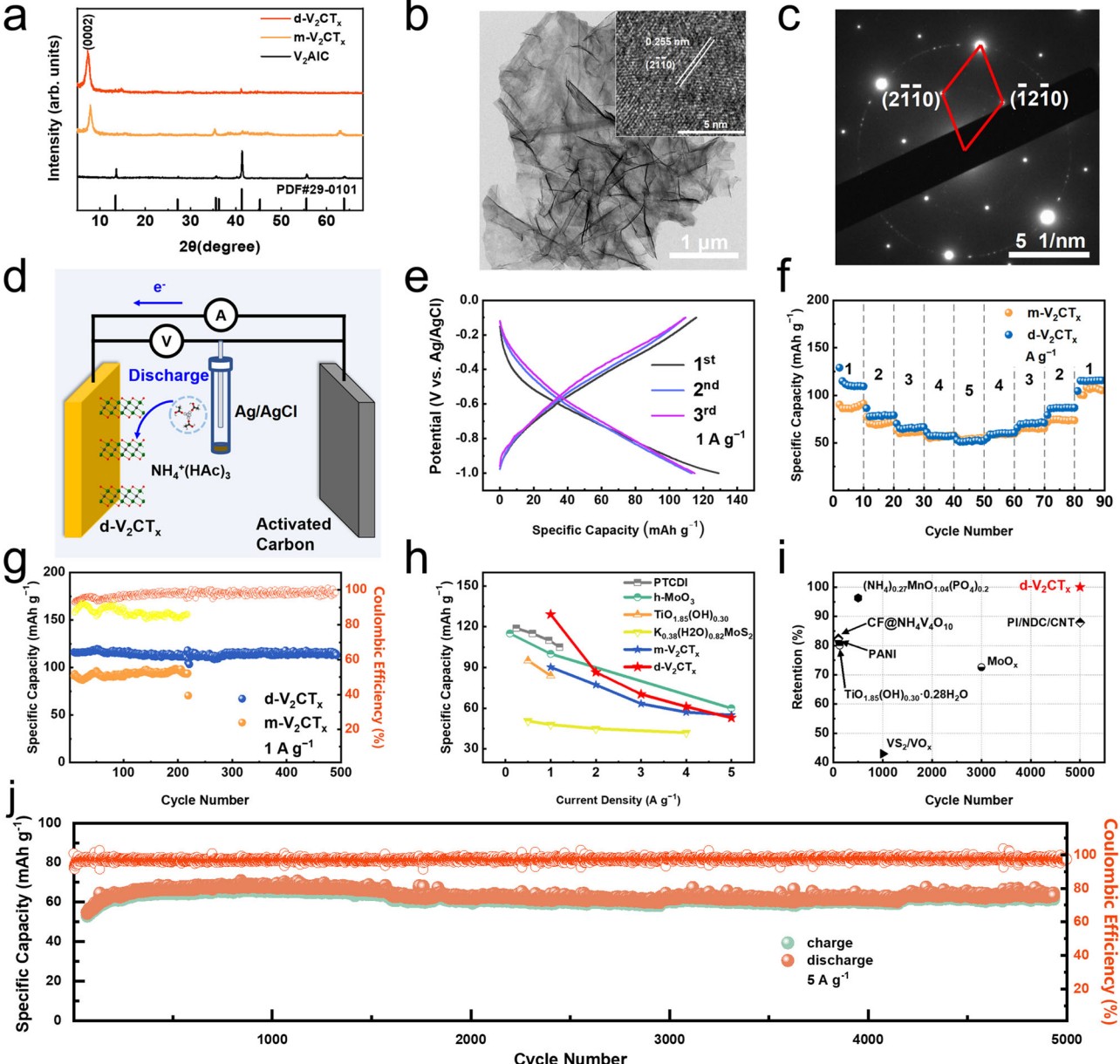

**Fig. 2 | Characterizations and half-battery performance of m-$V_2CT_x$, and d-$V_2CT_x$. a** XRD patterns of $V_2AlC$, m-$V_2CT_x$, and d-$V_2CT_x$. **b** TEM and HRTEM morphology of d-$V_2CT_x$. **c** Collected SAED patterns of d-$V_2CT_x$. **d** Scheme illustrating the structure of the three-electrode half-cell. **e** The 1st-3rd GCD voltage profiles. **f** Rate capability. **g** Cycling performance at $1\,A\,g^{-1}$. **h** Long-term cycling performance at $5\,A\,g^{-1}$ for 5000 cycles. **i** Rate capability comparison with literature data. **j** Cycling performance comparison with previously reported materials for aqueous AIBs.

the ($2\bar{1}\bar{1}0$) planes. Selected area electron diffraction (SAED) pattern of d-$V_2CT_x$ in Fig. 2c exhibited hexagonally arranged sharp diffraction spots which can be indexed to the [0001] zone axis pattern. The element mapping in Supplementary Fig. 4 illustrated a uniform distribution of composition elements including V, F, O, and C in the d-$V_2CT_x$ nanosheet, respectively. The X-ray photoelectron spectra (XPS) test further verified the presence of V, F, O, and C elements in the d-$V_2CT_x$ nanosheet (Supplementary Fig. 5a). In particular, there exited three pairs of peaks in the high-resolution spectrum of V element: the peak located at 517.3 eV belonged to the $V^{4+}$-$2p_{3/2}$, while those located at 515.8 eV and 514.3 eV were assigned to $V^{3+}$-$2p_{3/2}$ and $V^{2+}$-$2p_{3/2}$, respectively (Supplementary Fig. 5b). This result indicated the co-existence of $V^{4+}$, $V^{3+}$, $V^{2+}$ in the achieved d-$V_2CT_x$ MXene.

Subsequently, the ammonium ion storage performance of $V_2AlC$, m-$V_2CT_x$, and d-$V_2CT_x$ MXene was evaluated by Swagelok-type cells with a three-electrode configuration, as shown in Fig. 2d, with the MXene served as the working electrode, activated carbon as the counter electrode, 0.5 M $NH_4Ac$ as the electrolyte, and a saturated Ag/AgCl electrode acted as the reference electrode ($E = 0.197$ V vs. SHE), respectively. The galvanostatic discharge/charge (GCD) profiles in Fig. 2e demonstrated that the d-$V_2CT_x$ electrode delivered a capacity of 129.1 mAh $g^{-1}$ at a current density of 1 A $g^{-1}$ in the first discharging process, showing an initial coulombic efficiency of 89.78%. In comparison, the m-$V_2CT_x$ sample exhibited a relatively lower specific capacity of 90.2 mAh $g^{-1}$ at 1 A $g^{-1}$. In contrast, the pristine $V_2AlC$ sample exhibited almost no capacity less than 5 mAh $g^{-1}$ in the same condition (Supplementary Fig. 6), suggesting that the chemical etching process of $V_2AlC$ MAX was crucial to achieve a high capacity.

Note that the d-$V_2CT_x$ electrode exhibited typical pseudocapacitive discharging/charging behavior showing no apparent charging/discharging plateau according to the cyclic voltammetry (CV) curve given in Supplementary Fig. 7. Moreover, outstanding rate performance of the d-$V_2CT_x$ electrode can be concluded in Fig. 2f and Supplementary Fig. 8, with a high specific capacity of 115.9 mAh $g^{-1}$ at 1 A $g^{-1}$ and remaining 53 mAh $g^{-1}$ at 5 A $g^{-1}$. Remarkably, the rate capability of d-$V_2CT_x$ is not only much better than m-$V_2CT_x$, and also conventional anode materials recently developed for ammonium ion storage including PTCDI[5], $MoO_3$[9], $TiO_{1.85}(OH)_{0.30}$[30], $K_{0.38}(H_2O)_{0.82}MoS_2$[31], etc.

Long-term cycling stability is another essential factor in evaluating the performance of rechargeable batteries. Figure 2g showed the cycling performances of m-$V_2CT_x$ and d-$V_2CT_x$ electrodes at 1 A $g^{-1}$. In a 500-cycling test, the d-$V_2CT_x$ electrode delivers almost no capacity decay (114.0 mAh $g^{-1}$ in 1st and 113.0 mAh $g^{-1}$ in 500th cycle). In contrast, the m-$V_2CT_x$ electrode exhibited an inferior cycling performance (less than 200 cycles), indicating the larger specific surface area of d-$V_2CT_x$ is beneficial to achieve a stable specific capacity. The coulombic efficiency of d-$V_2CT_x$ (the orange line) remained above 95% after 500 cycles, which was obviously higher than that of m-$V_2CT_x$ (the yellow line). The morphology of the d-$V_2CT_x$ after 500 cycles is shown in the following SEM image (Supplementary Fig. 9). After cycles, the material still maintained a layered morphology, indicating its excellent stability. The long-cycling performance of the d-$V_2CT_x$ electrode was evaluated using a high current density (5 A $g^{-1}$) as shown in Fig. 2j. The plots indicate that the capacity reached its maximum value after 500 cycles, and the capacity retention stayed around 100% during the total 5000 cycles. To sum up, the electrochemical performance of the d-$V_2CT_x$ electrode achieved in this work is listed in Fig. 2h, i for a brief comparison with the previously reported ammonium ion storage materials[11,26,30,32,33], showing satisfactory rate performance and superior cycling stability to those known to date.

Figure 3a illustrates the working potential window of the d-$V_2CT_x$ as well as that of some other typical ammonium ion storage materials reported previously. Considering that little effort has been made to the study of anode materials for aqueous ammonium ion storage, it is satisfactory that the d-$V_2CT_x$ possesses a relatively low working

potential range within $-1 \sim -0.01$ V (vs. Ag/AgCl), making it promising anode candidate for AIBs. Subsequently, $Na_{0.6}MnO_2$ (NMO) was selected as the coupling cathode material in this work, owing to its suitable reaction potential and an ammonium ion storage capacity of 50 mAh $g^{-1}$ (Supplementary Fig. 10–11)[34,35], thereby a full cell was constructed with the schematic two-electrode configuration as illustrated in Fig. 3b. Typical CV profiles of this d-$V_2CT_x$/0.5 M $NH_4Ac$/NMO full cell at a current density of 1 mV $s^{-1}$ delivered a pair of anodic/cathodic peaks at $\approx 0.6/1.0$ V (Fig. 3c). GCD profiles of the as-constructed battery in Fig. 3d verified a specific capacity of 170 mAh $g^{-1}$ at 1 A $g^{-1}$ at the first cycle (based on the active mass of the anode). The full battery delivered a rate capacity of 99.1, 67.5, 56.9, 51.3, and 46.8 mAh $g^{-1}$ at 1, 2, 3, 4, and 5 A $g^{-1}$, respectively, and it recovered to 83.8 mAh $g^{-1}$ when the current density returned back to 1 A $g^{-1}$ (Fig. 3e). Long term cycling identified a cycling life of over 500 cycles at 1 A $g^{-1}$ with a specific capacity of 57.7 mAh $g^{-1}$ and remained a coulombic efficiency close to 100% (Fig. 3f). Our d-$V_2CT_x$/ /NMO full battery exhibited an energy density of 103.36 Wh $kg^{-1}$ at a power density of 1127.6 W $kg^{-1}$. Furthermore, electrochemical Impedance Spectroscopy (EIS) was employed to further reveal the fast ion diffusion in our system. Figure 3g depicted the Nyquist plots of our battery before and after 500 cycles. From the equivalent circuit used to fit the EIS data, the charge-transfer resistance (R2) reduced from 12.21 Ω to 6.9 Ω, indicating the small barrier for charge transfer inside the material, which was conducive to the rapid charging/discharging[36]. We also attempted to verify the practical application in flexible and wearable devices, a soft-packed battery was constructed in Fig. 3f, which easily drove a light-emitting diode (LED) indicator under varied bending states of 90°, demonstrating its excellent flexibility and potential application on wearable electronics.

## Acetate ions enhancement effect

The d-$V_2CT_x$ MXene was observed to exhibit the feature of pseudocapacitance capacity during the storage of aqueous ammonium ions. Surprisingly, such a pseudocapacitive behavior could only be achieved in the $NH_4Ac$ electrolyte. Figure 4a displays five CV curves (1 mV $s^{-1}$) in various electrolytes containing $NH_4^+$ cations. In particular, a pair of broad redox peaks at around $-0.70$ V and $-0.35$ V was clearly observed in 0.5 M $NH_4Ac$ electrolyte, which cannot be detected in other examined electrolytes ($NH_4Me$, $NH_4Cl$, $(NH_4)_2SO_4$, and $(NH_4)_2C_2O_4$) despite of the same $NH_4^+$ concentration. Consequently, a much higher capacity of 101.3 mAh $g^{-1}$ was obtained in $NH_4Ac$ electrolyte than that in other electrolytes (calculated to be 80.9, 68.9, 60.3, and 52.7 mAh $g^{-1}$ for $(NH_4)_2SO_4$, $NH_4Cl$, $(NH_4)_2C_2O_4$, and $NH_4Me$, respectively). We further investigated the influence of the cation species of the electrolyte salt using $Mg(Ac)_2$, $Zn(Ac)_2$, and LiAc, showing an apparent decline in the curve area with the absence of the redox peaks (Fig. 4b). In addition, the effect of $NH_4Ac$ on ammonium ion storage performance was evaluated by tailoring the salt concentration. Given that the selected $NH_4^+$-containing aqueous electrolytes were weakly acidic, we further examined the electrochemical properties by pH value regulation of the $NH_4Ac$ electrolyte. Specifically, three additional electrolytes with different pH values were prepared: 0.5 M HAc (pH = 1.5), 0.25 M mixed solution of HAc and $NH_4Ac$ (denoted as $NH_4Ac(H)$, pH = 3.7), and 0.25 M mixed solution of $NH_3\cdot H_2O$ and $NH_4Ac$ (denoted as $NH_4Ac(OH)$, pH = 9.8). According to the CV curves shown in Fig. 4c, a slight change of the electrochemical window can be observed, from $-1 \sim -0.01$ V (pH = 6.5) to $-0.9 \sim -0.01$ V (pH = 3.7) and further to $-0.9 \sim 0.2$ V (pH = 1.5) due to the ease of hydrogen evolution in acid solution. However, the absence of redox reaction peaks yielded to a remarkable deterioration in the capacity of d-$V_2CT_x$, demonstrating the importance of neutral (or nearly neutral) electrolytes in facilitating redox reaction. The effect of $NH_4Ac$ electrolyte concentration was tested finally, as summarized in Supplementary Fig. 12. A clear oxygen evolution reaction could be observed in 10 M and 20 M $NH_4Ac$ electrolytes when charged to $-0.1$ V. The specific capacity of d-$V_2CT_x$ in 20 M $NH_4Ac$

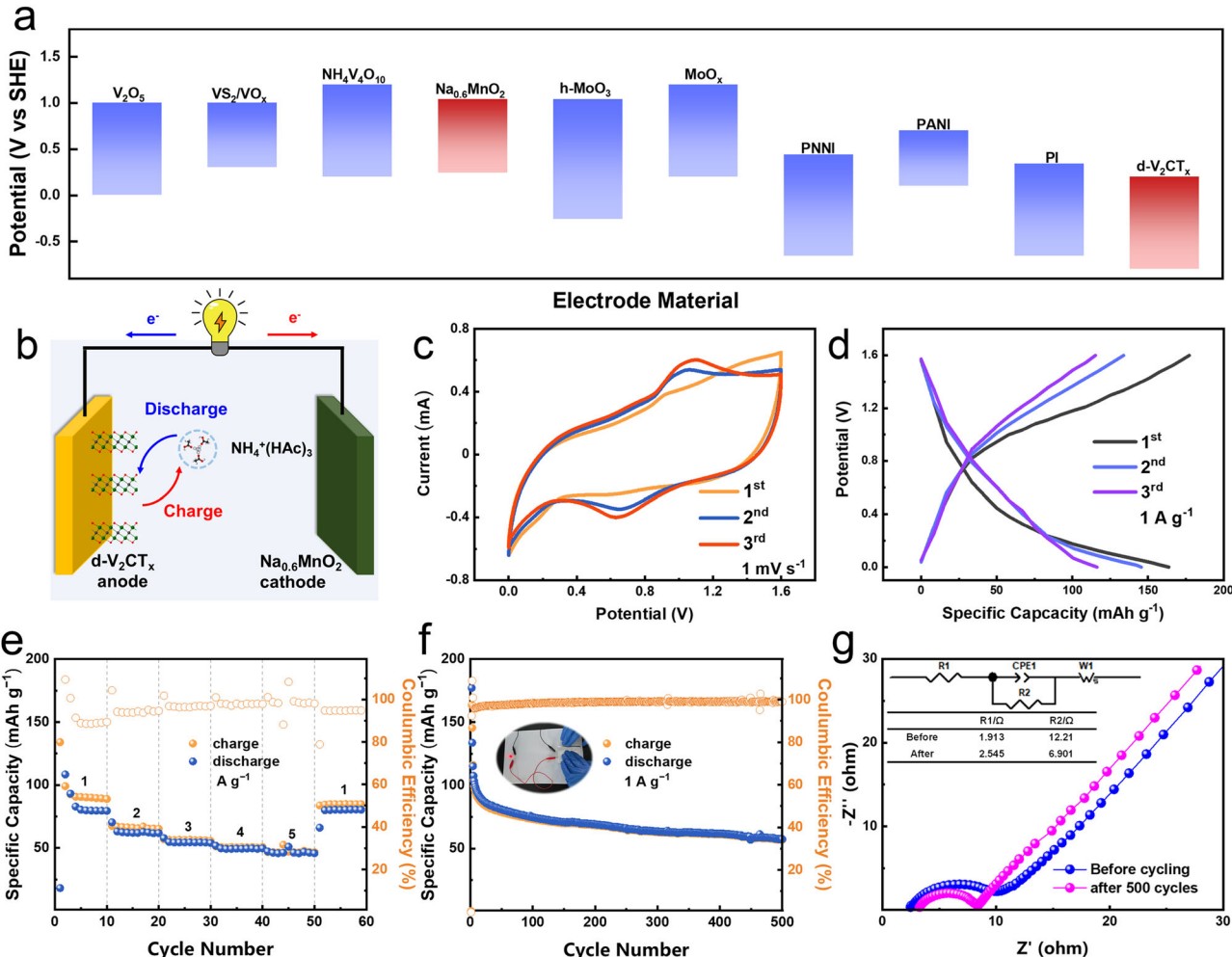

**Fig. 3 | Full cell construction. a** Comparison of working potential windows of $V_2CT_x$ MXene and other ammonium ion storage materials. **b** Scheme illustrating the structure of the d-$V_2CT_x$/0.5 M NH₄Ac/Na₀.₆MnO₂ full battery. **c** CV curves at 1 mV s⁻¹. **d** GCD curves at 1 A g⁻¹. **e** Rate capability at different current densities of 1-5 A g⁻¹. **f** Cycling performance at 1 A g⁻¹ with an insert showing a soft-packed battery driving an LED light. **g** EIS test before and after cycling.

electrolyte (88.6 mAh g⁻¹ at 1 A g⁻¹) was much inferior to that in 0.5 M NH₄Ac electrolyte (115.9 mAh g⁻¹ at 1 A g⁻¹), suggesting that a high concentration of NH₄Ac electrolyte may inhibit the electrochemical performance of $V_2CT_x$ (Supplementary Fig. 13).

To sum up, the specific capacity of the d-$V_2CT_x$ electrode in various salt electrolytes is collected in Fig. 4d, showing the optimal capacity of 101.3 mAh g⁻¹ uniquely achieved in the NH₄Ac electrolyte. It is worth noting that the key limitation of capacitive energy storage for both Faradaic (involving redox) and non-Faradaic (involving only electrostatic interactions) is its low capacity and unsatisfied energy density. Especially, in our case, benefited by this unique acetate ions enhancement effect, the specific capacity surpasses all of the as-reported capacitive-typed electrodes in ammonium-ion batteries up to date[9,37,38] (Fig. 4e).

The electrochemical kinetics were then elucidated by CV measurement at various scanning rates at 1, 2, 5, and 10 mV s⁻¹ in Fig. 4f. The contributions from diffusion-controlled and capacitive-controlled behaviors can be distinguished by analyzing the CV profiles according to the following equation (Eq. (1)) between the peak current ($i$) and the sweep rate ($v$)[39,40]:

$$i = av^b \tag{1}$$

whereas $b = 0.5$ stood for a diffusion-controlled process, and $b = 1$ stood for a capacitive-controlled process. In Fig. 4g, the $b$ value

of peak 1 and 2 was calculated to be 0.793 and 0.772, respectively, implying that the capacitive-controlled and diffusion-controlled processes synergistically dominate the energy storage process. The capacitive storage contributed to ≈58% as surface pseudocapacitance at 5 mV s⁻¹ (Fig. 4h). Furthermore, the capacitive-controlled process contributed 46%, 50%, 69% to the total capacity of d-$V_2CT_x$ at 1, 2, and 10 mV s⁻¹ (Supplementary Fig. 14). Similarly, the capacitive-controlled contributions of m-$V_2CT_x$ were calculated to be 34%, 40%, 51%, 63%, respectively (Supplementary Fig. 15). These values were lower than those observed in d-$V_2CT_x$, indicating an enhanced contribution from increased surface area by delamination treatment[41,42]. Also, the capacitive storage contributions in 0.25 M (NH₄)₂SO₄ electrolyte were calculated using the same measurement (Supplementary Fig. 16a). The capacitive-controlled process contributes 43%, 45%, 54%, and 69% to the total capacity of d-$V_2CT_x$ at 1, 2, 5, and 10 mV s⁻¹, respectively (Supplementary Fig. 16b). Based on these data, the capacitive-controlled capacity was evaluated to be 46.5 mAh g⁻¹ in 0.5 M NH₄Ac electrolyte (Supplementary Fig. 17), which was higher than the 35.3 mAh g⁻¹ in 0.25 M (NH₄)₂SO₄ electrolyte. All of the results confirm that a unique pseudocapacitive reaction happens in the NH₄Ac electrolyte and enhances the capacitive-controlled process. The redox reaction which significantly contributed to the capacity of d-$V_2CT_x$ in NH₄Ac electrolyte was attractive, the mechanism of which would be discussed in our following sections. The ex-situ XPS characterization and in-situ electrochemical quartz crystal microbalance (EQCM)

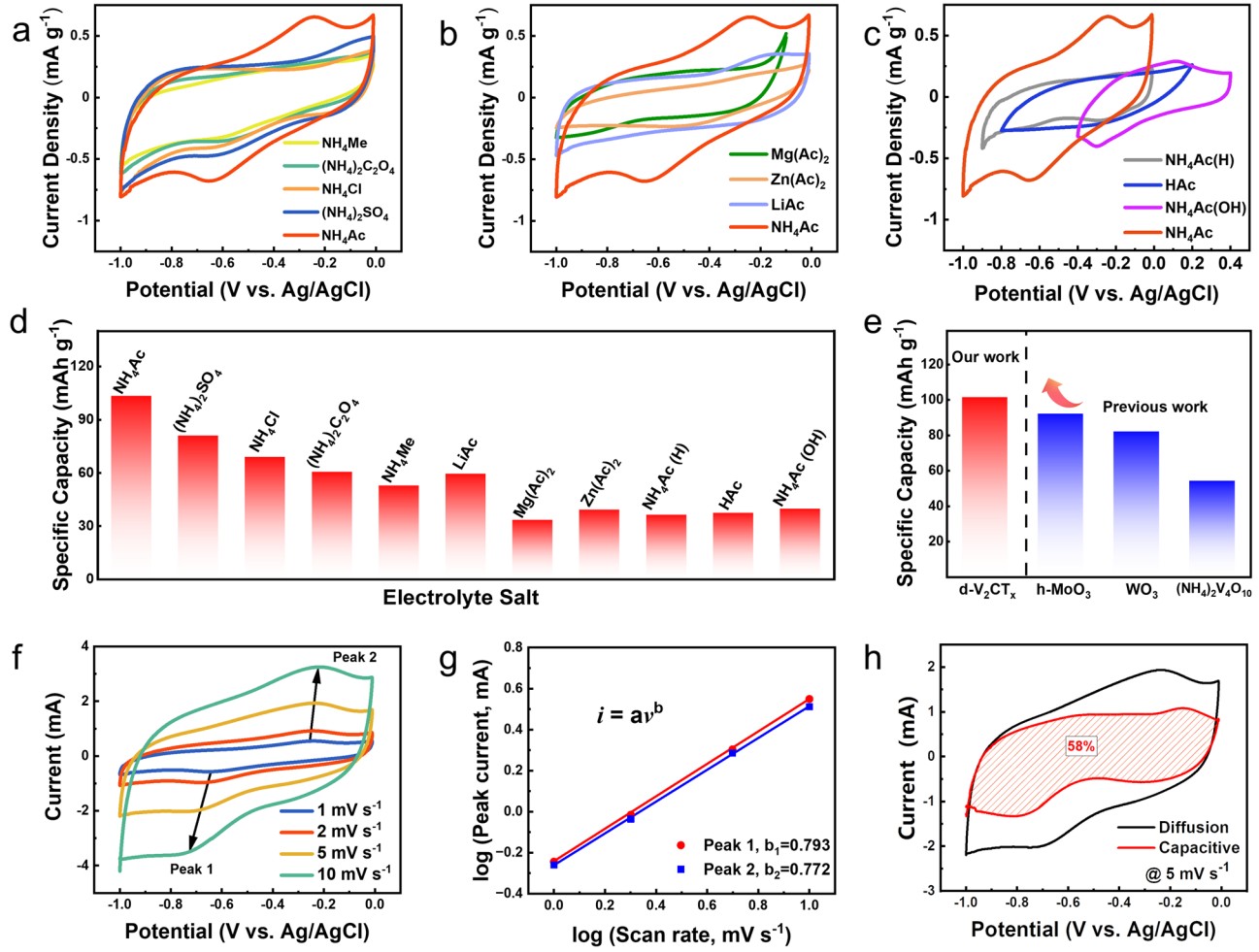

**Fig. 4 | Electrolyte dependent ammonium ion storage performance. a** CV curves of d-V$_2$CT$_x$ in 0.5 M NH$_4^+$ electrolytes at 1 mV s$^{-1}$. **b** CV curves of d-V$_2$CT$_x$ in 0.5 M Ac$^-$ electrolytes at 1 mV s$^{-1}$. **c** CV curves of d-V$_2$CT$_x$ in 0.5 M NH$_4$Ac electrolytes with pH regulation at 1 mV s$^{-1}$. **d** Summary of the specific capacity values of d-V$_2$CT$_x$ in different electrolytes. **e** Capability comparison with other pseudocapacitive ammonium ion storage materials. **f** CV curves of d-V$_2$CT$_x$ at different scanning rates at 1, 2, 5, and 10 mV s$^{-1}$ in 0.5 M NH$_4$Ac showing redox peaks. **g** log $i$ versus log $v$ plotted according to redox peaks. **h** capacitive-controlled contribution calculated at 5 mV s$^{-1}$.

measurement was carried out. The ex-situ XPS results clearly identified the evolution of chemical composition as well as the valence state of d-V$_2$CT$_x$ during the Faraday process (Supplementary Fig. 18). The active materials in the working electrode using a three-electrode configuration after five discharging/charging cycles were selected due to its coulombic efficiency approached 100% at this time. In the high-resolution V $2p_{3/2}$ XPS spectrum, it was notable that the valence state of V at the fully charged state was composed of +4 and +3, which is different from the initial state of V$_2$CT$_x$ (+4, +3, and +2)[43]. During the discharging process, the characteristic peak of V$^{4+}$ (517.7 eV) was evident at Point A (−0.1 V) but disappeared at Point B (−0.7 V). Instead, the peak of V$^{3+}$ (516.5 eV) increased significantly, accompanied by the appearance of the V$^{2+}$ peak (513.9 eV). In the charging process, the valence state of V remained constant at Point C (−1 V). While charged to Point D (−0.15 V), the V$^{4+}$ peak rose again to a high level. Furthermore, O 1 $s$ XPS spectra indicated a reversible transformation of O valence, in which the characteristic peak (530.4 eV) of the V−O bond decreased at −0.7 V in the discharging process, and recovered at −0.15 V in the charging process. The other peak at 532.0 eV represented the V-O···HN which increased at −0.7 V in the discharging process. Otherwise, ex-situ XRD was employed to examine the changes in layer spacing during the charging and discharging process, as shown in Supplementary Fig. 19. The diffraction peak representing the V$_2$CT$_x$ (0002) facet at fully discharged state showed almost no change compared to that at

the fully charged state, suggesting that the interlayer spacing of the d-V$_2$CT$_x$ electrode generally stayed unchanged during the whole electrochemical process.

In-situ EQCM measurement provides an effective approach to understanding the deposition/dissolution process on the electrode surface thereby disclosing the charge storage mechanism. Subsequently, a three-electrode cell utilizing quartz microcrystals as the current collector was constructed as shown in Supplementary Fig. 20. Figure 5a presents the CV curve with a potential window of −1.0 to −0.01 V versus Ag/AgCl of the EQCM cell at 2 mV s$^{-1}$. The curve appeared to be relatively smooth from point A to B, then one peak rose between −0.5 ~ −0.7 V during the discharging process. According to Sauerbrey's equation and Faraday equation (see in Methods), the apparent molar mass of interacted ions ($M_w$, g mol$^{-1}$ e$^{-1}$) correlated with the mass change per coulomb ($\Delta m/\Delta Q$) which can be calculated from the slope in Fig. 5b. Accordingly, the mass increased slowly in the range of −0.01 V ~ −0.6 V (from point A to B), which were estimated with a $M_w$ of 18 g mol$^{-1}$ e$^{-1}$. Normally, there are two possible interactive species (NH$_4^+$, Ac$^-$) that may be associated with 0.5 M NH$_4$Ac electrolyte. Note that the as-detected value of 18 g mol$^{-1}$ e$^{-1}$ is very close to the molecular weight of NH$_4^+$ group, such a mass increase can be attributed to the adsorption of NH$_4^+$ group on MXene surface. In −0.7 V ~ −1.0 V range (from point C to D), the mass increased until the end of the discharge period with a $M_w$ of 196 g mol$^{-1}$ e$^{-1}$, which was

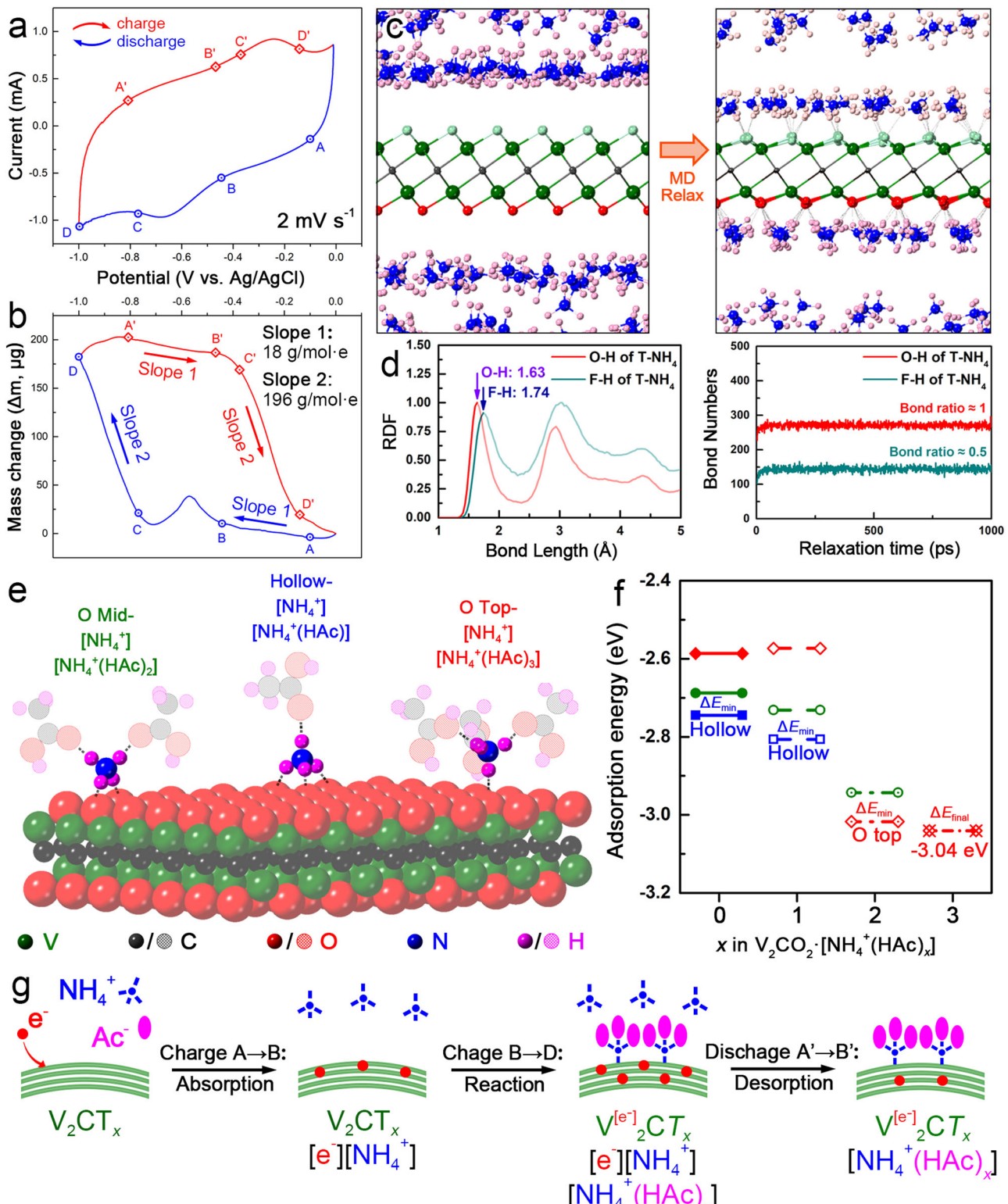

**Fig. 5 | Pseudocapacitive origin of V$_2$CT$_x$ in NH$_4$Ac electrolyte. a** CV curves of d-V$_2$CT$_x$ in 0.5 M NH$_4$Ac electrolytes in the EQCM cell. **b** mass change of d-V$_2$CT$_x$ in a single discharging/charging process. **c** adsorption configuration analysis obtained by MD simulation (**d**) the radial distribution function (RDF) and bond numbers between V$_2$CT$_x$ (T = -F, -O) and [NH$_4^+$] ions; (**d**) adsorption configurations obtained

by DFT calculation showing the interaction between V$_2$CO$_2$ and [NH$_4^+$], [NH$_4^+$(HAc)$_x$] groups. **e** the adsorption energy calculation results on the surface of V$_2$CO$_2$. **f** DOS analysis of V element in V$_2$CO$_2$ and V$_2$CO$_2$[NH$_4^+$(HAc)$_3$]. **g** scheme displaying the adsorption of [NH$_4^+$] ions on V$_2$CO$_2$ and reaction of [NH$_4^+$(HAc)$_x$] groups with V$_2$CO$_2$.

close to the mass sum of one $NH_4^+$ and three HAc molecules ($[NH_4^+(HAc)_3]$). Conversely, there occurred a two-stage mass drop during the charging process. First, the mass decreased by $18\,g\,mol^{-1}\,e^{-1}$ from A′ to B′, and the $M_w$ suddenly changed to $196\,g\,mol^{-1}\,e^{-1}$ from C′ to D′ when the potential reached −0.4 V. Combined with the CV curve in Fig. 5a, the large $M_w$ value of $196\,g\,mol^{-1}\,e^{-1}$ occurred in the process of the redox reaction, making it rational to be attributed to the redox reaction between the electrode and electrolyte.

In this work, both DFT calculation and molecular dynamics (MD) simulation were performed to better understand the ion/electron transfer and pseudocapacitive origin in the NH₄Ac electrolyte. The $[NH_4^+]$ ion possessing a unique tetrahedral-shaped multipole was capable of rotating to maintain a subset of coordinated H-bonds with four oxygen atoms from both electrode and electrolyte[6]. After a full relaxation in both DFT calculation and MD simulation, the possible adsorption configurations of $[NH_4^+]$ ions and $[NH_4^+(HAc)_x]$ groups on the surface of $V_2CO_2$ were collected and displayed in Fig. 5c–e, Supplementary Fig. 21–22, which can be generally divided into three types according to the number of hydrogen bonds: the "O top-$[NH_4^+]$" (with the central N located above O termination, labeled by red) bonds with 1×O termination from $V_2CO_2$ which was capable of holding 1 - 3×HAc; the "O mid-$[NH_4^+]$" (with the central N located at the middle of two O terminations, labeled by green) bonds with 2×O terminations from $V_2CO_2$ which was capable of grasping 1 - 2×HAc; the "Hollow-$[NH_4^+]$" (with the central N located at the hollow of three O terminations, labeled by blue) bonds with 3×O terminations which was capable of dangling 1×HAc. Moreover, the MD simulation in Fig. 5c–d, Supplementary Fig. 23–24 clearly demonstrated that a much higher coverage rate of the $[NH_4^+]$ ions can be obtained on the surface of the $V_2CO_2$ model than $V_2CF_2$ and $V_2C(OH)_2$. This result indicates that $V_2CT_x$ having rich -O terminations can achieve high capacitance. A detailed DFT calculation of the adsorption energy was displayed in Fig. 5e. The Hollow-$[NH_4^+]$ model possessed the lowest formation energy (−2.74 eV) among all three possible ion adsorption configurations, which was energetically more favorable than the models of O mid-$[NH_4^+]$ (−2.69 eV) and O top-$[NH_4^+]$ (−2.59 eV). Notably, there exists an extra Coulomb interaction between $V_2CT_x$ and $NH_4^+$, from the negative surface potential of $V_2CT_x$ and the positive charge of $NH_4^+$ which is neglected in the DFT simulation. In fact, the adsorption of $NH_4^+$ ions on the surface of $V_2CT_x$ in the experiment situation was more feasible than in the theoretical situation. As a consequence, the electrical double-layer capacity (EDLC) occurred in the first stage of discharging process, corresponding to the $18\,g\,mol^{-1}\,e^{-1}$ mass gaining (A → B) observed in the EQCM test. With the saturation of $[NH_4^+]$ adsorption (strongly dependent on the surface area of $V_2CT_x$), the energy storage mechanism turned to pseudocapacity (PC) where the valence state of V changed with the occurrence of reaction between $V_2CT_x$ and $[NH_4^+(HAc)_x]$ groups. It has been widely accepted that the alternation of surface terminations, generally from -O to -OH, played a significant role in the electron storage mechanism of MXene electrodes. While in the aqueous ammonium ion system utilizing a neutral (or nearly neutral) NH₄Ac electrolyte, the lean of proton made such a termination alternation difficult. As a consequence, the reaction between $V_2CT_x$ and $[NH_4^+(HAc)_x]$ facilitated the redox reaction, by providing -O termination with a $[NH_4^+]$ which acted as a pseudo-proton. The optimal group adsorption configurations of $[NH_4^+(HAc)_x]$ were achieved at Hollow-$[NH_4^+(HAc)]$ for 1×HAc (−2.81 eV), O top-$[NH_4^+(HAc)_2]$ for 2×HAc (−3.02 eV) and O top-$[NH_4^+(HAc)_3]$ 3×HAc (−3.04 eV), respectively (Fig. 5f). The configurations of groups indicate a necessary rotation of $[NH_4^+]$ ions with the rising number of HAc molecules. As a consequence, the surface reaction of $[NH_4^+(HAc)_x]$ group mainly contributed to the pseudocapacity in the second stage of discharging process, corresponding to the $196\,g\,mol^{-1}\,e^{-1}$ mass gaining (C → D) observed in the EQCM test, and the rotation of $[NH_4^+]$ ions as well the alternation of surface termination of $V_2CT_x$ led to a surface group

redistribution which accounted for the mass fluctuation (B → C) observed in the EQCM curve. The change in the valence state of V elements posed a great impact on the bonding properties of $V_2CT_x$. Supplementary Fig. 25 plotted the density of states (DOS) of V-$d$ orbital in $V_2CO_2$ and $V_2CO_2[NH_4^+(HAc)_3]$, respectively. It can be seen that the V3$d$-O2$p$ hybridization mainly contributed to the orbitals around Fermi level. With the introduction of $[NH_4^+(HAc)_3]$ group on the surface, general segregation of wave-functions around Fermi level can be observed: the bonding orbitals (BO) below Fermi level slightly shifted towards low energy (the area center of which moves from −3.60 eV to −3.64 eV), indicating a possible strengthening effect on V-O bonding, while the anti-bonding orbitals (ABO) above Fermi level slightly shifted towards high energy (the area center of which moved from 0.92 eV to 1.14 eV, implying a possible enhancing capability to accommodate excited electrons, especially with the presence of an extra orbit at 2.15 eV. The charge density difference plotted in Supplementary Fig. 26 demonstrated that surface terminations of $V_2CT_x$ played a significant role in realizing the charge transfer between V element and $[NH_4^+(HAc)_3]$ group: according to the Bader analysis result, the valence state of V decreased from +1.78 (positive for losing electrons) to +1.69 with the presence of one $[NH_4^+(HAc)_3]$ group on the surface of $V_2CO_2$ (a $3 \times 3 \times 1$ supercell), which align with the conclusions from ex-situ XPS characterization (Supplementary Fig. 18).

Finally, the ammonium ion storage mechanism of the $V_2CT_x$ electrode in the NH₄Ac electrolyte was summarized in Fig. 5g. In the first stage of discharging process, the electrons were stored on the surface of $V_2CT_x$ electrode, dominated by the EDLC mechanism accompanied by the electrostatic adsorption of $[NH_4^+]$ ions. In the second stage of discharging process, the electrons were stored on the V site of $V_2CT_x$ electrode, yielding to the alternation of valence state companied by the coordination of surface terminations (from -O to -O···HN), relating with the surface reaction between $V_2CT_x$ and $[NH_4^+(HAc)_x]$ groups, which triggered the as-observed notable pseudocapacitive behavior. The whole electrochemistry reaction equation (Eqs. (2–5)) can be illustrated as follows:

(1)discharging process:

$$V2CT_x + NH_4^+ + e^- \rightarrow V2CT_x \cdot (e^-) \cdot [NH_4^+] \, (adsorption, EDLC) \quad (2)$$

$$V2CT_x + NH_4^+(HAc)_3 + e^- \rightarrow (V,e^-)_2CT_x \cdot [NH_4^+(HAc)_3] \, (reduction, PC) \quad (3)$$

(2)charging process:

$$V2CT_x \cdot (e^-) \cdot [NH_4^+] \rightarrow V2CT_x + NH_4^+ + e^- \, (desorption) \quad (4)$$

$$(V, e^-)2CT_x \cdot [NH4^+(HAc)_3] \rightarrow V2CT_x + NH_4^+(HAc)_3 + e^- \, (oxidation, PC) \quad (5)$$

## The generality in aqueous ammonium ion storage

The generality of this acetate ion enhancement effect on pseudocapacitive capacity was further confirmed in 1T-MoS₂-based ammonium-ion battery using a Swagelok-type cell with the layered 1T-MoS₂ as the working electrode (Fig. 6a), activated carbon as the counter electrode, and a saturated Ag/AgCl electrode acted as the reference electrode, respectively. 0.5 M NH₄Ac electrolyte and 0.25 M $(NH_4)_2SO_4$ electrolyte were selected as two different electrolyte systems with the same $NH_4^+$ concentration. The GCD profiles in Fig. 6b indicate the apparent discrepancy between these two different electrolyte systems. The specific discharge capacity at $1\,A\,g^{-1}$ is $78.0\,mAh\,g^{-1}$ in 0.5 M NH₄Ac electrolyte, which is much larger than that in 0.25 M $(NH_4)_2SO_4$ electrolyte ($55.0\,mAh\,g^{-1}$). The cyclability of the 1T-MoS₂ anode in NH₄Ac electrolyte was also superior to that in $(NH_4)_2SO_4$ electrolyte (Supplementary Fig. 27). Furthermore, the CV curve area obtained in NH₄Ac

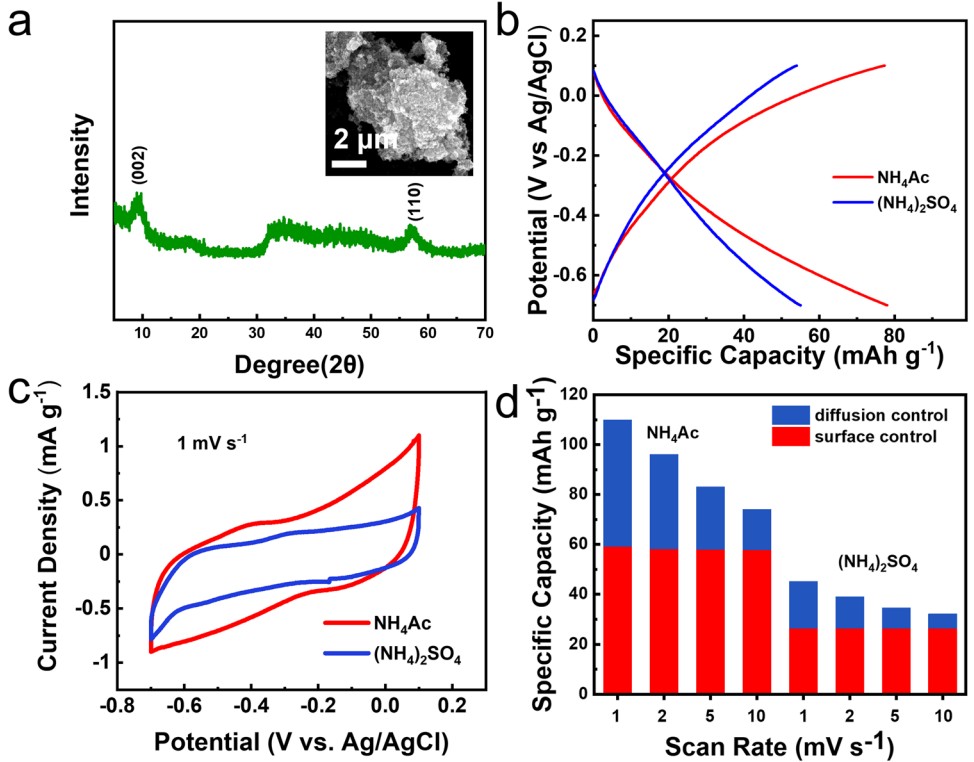

**Fig. 6 | The generality of the acetate ion enhancement effect in 1T-MoS₂-based ammonium-ion battery. a** XRD pattern and SEM image of 1T-MoS₂. **b** GCD profiles of 1T-MoS₂ in 0.5 M NH₄Ac and 0.25 M (NH₄)₂SO₄ at 1 A g⁻¹. **c** CV curves achieved in these two electrolytes at 1 mV s⁻¹. **d** Contributions of 1T-MoS₂ in these two electrolytes at varied scan rates.

electrolyte was remarkably larger than that in $(NH_4)_2SO_4$ electrolyte at 1 mV s⁻¹ (Fig. 6c). The reaction process was further assessed by CV at different sweep rates (Supplementary Fig. 28). According to Eq. (1), the surface-controlled capacity was evaluated to be 58.2 mAh g⁻¹ in 0.5 M $NH_4Ac$ electrolyte (Fig. 6d), which was obviously higher than the 37.6 mAh g⁻¹ in 0.25 M $(NH_4)_2SO_4$ electrolyte. All of the aforementioned results confirmed the acetate ion enhancement effect is also striking in the 1T-MoS₂-based ammonium-ion battery.

## Discussion

In summary, our study demonstrated that $V_2CT_x$ MXene is a robust anode material for high-performance aqueous AIBs by both theoretical calculation and experimental data. Pseudocapacitive ammonium ion storage behavior has been observed in $V_2CT_x$ MXene for aqueous AIBs, in which an optimized $NH_4Ac$ electrolyte plays a crucial role in this pseudocapacitive behavior. In-situ EQCM measurement reveals a two-step electrochemical process: the first one is the electrostatic adsorption/deposition of $NH_4^+$ on the MXene surface. In the second process, a redox reaction occurs between $d-V_2CT_x$ and $[NH_4^+(HAc)_3]$ group, during which process the central $NH_4^+$ acts as a pseudo-proton facilitating the termination alternation from -O to -O···HN, thereby realizing the variation of V valence state. Benefited from the high reversibility of this pseudocapacitive behavior, the $d-V_2CT_x$ MXene exhibits a high reversible capacity (115.9 mAh g⁻¹ at 1 A g⁻¹) and an excellent cycling stability (no decay for 5000 cycles at 5 A g⁻¹). This specific capacity surpasses all of the as-reported capacitive-typed electrodes in ammonium-ion battery up to date. The generality of this acetate ion enhancement effect on pseudocapacitive capacity is further confirmed in the MoS₂-based ammonium-ion battery system. Our work opens a new door to realize high capacity on sustainable ammonium ion storage by the acetate ion enhancement effect. It also makes a breakthrough in the capacity limitation of capacitive energy storage for both Faradaic (involving redox) and non-Faradaic (involving only electrostatic interactions) types.

## Methods

### Preparation of materials

The $V_2AlC$ MAX was prepared is by pressureless sintering in an Ar atmosphere. Vanadium (99.5%), aluminum (99.5%), and graphite (99%) were mixed in a 2:1.1:0.9 atomic ratio. The precursors were shaken in a powder mixer for 12 h to achieve a homogenous state. The mixture was transferred into alumina crucibles and placed into a high-temperature tube furnace. After replacing the atmosphere with ultrahigh purity Ar gas, the furnace is heated to 1550 °C at a rate of 5 °C min⁻¹, held for 2 h, and then cooled to room temperature at a rate of 5 °C min⁻¹. The $V_2AlC$ MAX phase was crushed using a mortar and pestle and then sieved to the desired particle size (>300 mesh).

$m-V_2CT_x$ was prepared by in-situ HF etching method as reported. The mixture of LiF/HCl was prepared by dissolving 2 g of LiF in 20 ml of 12 M HCl. Then 2 g $V_2AlC$ was immersed into the solution at 90 °C for 72 h. After etching, the resultant suspension was washed with deionized water and 75% alcohol several times. After that, the product was dried in a vacuum oven at 60 °C for 12 h. In this work, the $V_2AlC$ phase was also treated with 40% HF for 72 h at room temperature to compare its ammonium-ion storage performance.

$d-V_2CT_x$ was obtained by tetramethylammonium hydroxide (TMAOH) treatment as reported by Yury. In detail, 1 g dried $m-V_2CT_x$ powder was added to 20 ml of a 10 wt% TMAOH solution in water, and the solution was stirred at room temperature for 6 h. After stirring, the solution was centrifuged at 1350 × g. for 30 min and the supernatant was decanted. Then 20 ml deionized water was added to the $V_2CT_x$ MXene. the $d-V_2CT_x$ solution was obtained after being hand-shaken for 1–2 min and centrifuged at 700 × g. for 10 min. Then the $d-V_2CT_x$ powder was obtained by vacuum filtering.

1T-MoS$_2$ was prepared by hydrothermal method. In brief, 300 mg of ammonium molybdate (99.9%), 360 mg of thioacetamide (99.9%), 1.20 g of urea (99.9%), and 6.40 g of lithium sulfate (99.9%) were added to 100 ml of deionized water with magnetic stirring. Then, the mixture was transferred to a Teflon-lined stainless-steel autoclave and held at 180 C for 18 h. Finally, 1T-MoS$_2$ was obtained after rinsing with deionized water and ethanol several times.

## Materials characterizations

The crystal structure characteristics were studied by X-ray diffraction (XRD, Bruker D8 X-ray diffractor with Cu Kα radiation (λ = 1.5406 Å)). The morphology and structure of samples were characterized using scanning electron microscopy (SEM, Sirion 200) and transmission electron microscopy (TEM, FEI Talos F200X). Corresponding selected electron diffraction (SAED) and energy-dispersive spectrum (EDS-mapping) tests were carried out accompanied by the TEM measurement. The elemental composition and chemical state of samples were measured by X-ray photoelectron spectrum (XPS, Thermo Scientific Escalab 250Xi). The nitrogen adsorption and desorption isotherms were recorded by a Quanta Autosorb-IQ2 analyser. The samples for BET testing were degassed at 120 °C for 12 h in a vacuum and then transferred to the analyzer for testing.

## Electrochemical characterizations

V$_2$CT$_x$ MXene, acetylene black, and polyvinylidene fluoride (PVDF) with a weight ratio of 7:2:1 were mixed in N-methylpyrrolidone (NMP) with stirring. The obtained homogeneous slurry was sprayed on carbon paper and vacuum-dried at 80 °C for 12 h. The obtained carbon paper was punched into small disks with a diameter of 1.0 cm with a V$_2$CT$_x$ mass loading of -1.3 mg cm$^{-2}$. The electrode had an area of 0.785 cm² and a thickness of 10 μm. The separator had an area of 1.13 cm² and a thickness of 20 μm. The electrochemical measurements of the V$_2$CT$_x$ MXene were carried out in the three-electrode Swagelok cell in (NH$_4$)$_2$SO$_4$, NH$_4$Cl, NH$_4$Me, (NH$_4$)$_2$C$_2$O$_4$ and NH$_4$Ac electrolytes, Ag/AgCl electrode was set as reference electrode. The activated carbon served as the counter electrode, which mixed acetylene black and PVDF based on a mass ratio of 7:2:1. The cyclic voltammetry (CV) and electrochemical impedance spectroscopy (EIS) tests were conducted on an electrochemical workstation (CHI660E). The galvanostatic charge-discharge (GCD) and long-term cycling tests were recorded on a LAND battery test system (CT3001A). The electrochemical potential window is −1.0 ~ −0.01 V (vs. Ag/AgCl), and the GCD current density ranges from 1 - 5 A g$^{-1}$. The performances of the full cell were tested in CR-2032 coin-type cells which were assembled using NH$_4$Ac electrolytes, glass fiber membranes (Whatman, GF/D), and Na$_{0.6}$MnO$_2$ as the electrolyte, separator, and cathode electrode, respectively. In-situ EQCM measurement was carried out in a three-electrode cell using quartz microcrystals (QCM200, Stanford Research Systems, Inc.) as the current collector. The CV and GCD measurements were conducted using an electrochemical workstation (Biologic SP-150). During the electrochemical process, ions and molecules interact with the V$_2$CT$_x$ MXene electrode causing the resonance frequency change. The EQCM response correlates with the mass changes of the electrode due to ions and/or solvent molecules interactions according to the following Sauerbrey's equation (Eq. (6)):

$$\Delta f = - C_f \cdot \Delta m \qquad (6)$$

where $\Delta f$ and $\Delta m$ represent frequency and mass change, respectively. C$_f$ is the sensitivity factor for the crystal (56.6 Hz μg$^{-1}$ cm²). By applying a constant current to the EQCM cell, the apparent molar mass of interacted ions ($M_w$, g (mol e)$^{-1}$) can be calculated according to

Faraday's law in the equation (Eq. (7)):

$$M_w = \frac{F \Delta m}{\Delta Q} \qquad (7)$$

where $\Delta Q$ is the charge passed through the electrode in coulomb, $F$ is the Faraday constant (96485 C mol$^{-1}$).

## Computational method

First-principles calculation together with molecular dynamics simulation was performed in this work for a comprehensive understanding of the adsorption/desorption behaviors on the V$_2$CT$_x$ MXene electrode. The classical molecular dynamics (MD) simulations were carried out using the LAMMPS code in this work, using the Universal force field[44] and CVFF[45] force field. A large supercell with the size of 12 × 12 × 1 was built in a 35 × 35 × 35 Å³ box, with a chemical formula of V$_{288}$C$_{144}$T$_{288}$ (T = -F, -O, -OH). Thereafter, the Metropolis Monte Carlo method was employed to introduce 386 NH$_4^+$ ions or 161 NH$_4$Ac group. All the simulations were carried out with a time step of 1 fs. All the systems were relaxed for 1000 ps under isothermal-isobaric ensemble. The temperature was controlled at 300 K using the Nosé-Hoover thermostat with a temperature damping parameter of 0.1 ps and the pressure was also controlled using the Nosé-Hoover barostat with a pressure damping parameter of 1 ps. Thereafter, the radial distribution functions (RDFs) were statistically calculated on the last 500 ps of relaxation. Pertinent force field parameters are presented in Supplementary Table 1.

The Vienna Ab initio Simulation Package (VASP) was adopted, with the Projector Augmented Wave (PAW) method applied to solve the Kohn-Sham equations, and the functional of Generalized Gradient Approximation constructed by Perdew-Burke-Ernzerhof (GGA-PBE) adopted as exchange-correlation energy functional. The two-dimensional V$_2$CT$_x$ (T = -F, -O and -OH) models were built for the simulation, and the working function was calculated according to $\Phi = E_{Fermi} - E_{vacuum}$, which evaluates the energy required to activate an electron from Fermi level to vacuum. A supercell size of 3 × 3 ×1 of V$_2$CO$_2$ was adopted for simulating its interaction with [NH$_4^+$] ion and [NH$_4^+$(HAc)$_x$] groups, with adsorption energy calculated using an implicit solvation model (VASPsol) according to the equation (Eq. (8)):

$$V_2CO_2 \cdot e^- + NH_4^+(HAc)_{x(aq)} \xrightarrow{\text{adsorption}} V_2CO_2 \cdot NH_4(HAc)_x \qquad (8)$$

All the models studied in this work were geometrically optimized using a cutoff energy of 480 eV with a $k$-mesh of 3 × 3 × 1, during which process the convergence criteria were set to be 10$^{-4}$ eV for energy and 10$^{-2}$ eV/Angst for force. The density of states of studied models was computed using highly accurate parameters, at a dense $k$-mesh of 15 × 15 × 5 and smearing width of 0.05 eV.

## Reporting summary

Further information on research design is available in the Nature Portfolio Reporting Summary linked to this article.

# Data availability

All relevant data are available from the authors, and requests for datasets should be addressed to L.F.H. or Z.S.

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

## Acknowledgements

This work was financially supported by the National Key Research and Development Program of China (Grant No. 2021YFB2400400), the National Natural Science Foundation of China (Nos. 52171203, 52371214, 52250010, 52302224), the Natural Science Foundation of Jiangsu Province (Grants No. BK20211516), the Fundamental Research Funds for the Central Universities (2242023K5001), and the Project on Carbon Emission Peak and Neutrality of Jiangsu Province (BE2022031-4).

## Author contributions

L.F.H., and Z.S. developed the material design and experimental study. Z.B., Q.L. performed materials synthesis, battery assembly, and performance measurement. Z.B. and Q.L. performed SEM, TEM, XRD, and XPS. C.L. performed DFT calculations. W.L. and Y.Z. performed MD simulations. L.B.D., H.T., F.Y., L.P., and Y.P.W. helped to discuss the experimental result. L.F.H., and Z.S. supervised the project. All authors contributed to the discussion of the results and writing the paper.

## Competing interests

The authors declare no competing interests.
