## [Peer Review File · Nature Communications]

An acetate electrolyte for enhanced pseudocapacitive capacity in aqueous ammonium ion batteriesREVIEWER COMMENTS

Reviewer #1 (Remarks to the Author):

In this manuscript, the authors reported a novel, pseudocapacitive typed energy storage behavior with unique acetate ion enhancement effect in V₂CT_x MXene. It is impressive that its specific capacity surpasses all of the capacitive-typed electrodes in ammonium-ion battery up to date. Furthermore, the insight on acetate ion enhancement effect shows general interests on electrolyte design for aqueous ion battery. Therefore, I consider it is a nice paper and recommend its publication after some minor revision:

1. Is the conclusion applicable to other MXenes, such as the common Ti₃C₂T_x?
2. It is required to give more information about BET results. Also, the equipment information and the preparation of the BET samples should be provided;
3. In Supplementary Fig 4b, V exhibits three valence states, why only two valence states exist in ex-situ XPS (Supplementary Fig 13)?
4. Please supplement the material characterization after cycling to further demonstrate its stability.
5. In Fig 3a, has the author attempted to match a full cell with positive electrode materials other than NMO?
6. Please give the capacitive-controlled contribution in other electrolytes and compare them with that in NH₄Ac.
7. Please provide the specific capacity of MXene in high concentration NH₄Ac electrolyte.

Reviewer #2 (Remarks to the Author):

Ammonium ion batteries represent an intriguing prospect in the realm of electrochemical energy storage technologies for the future. However, a critical bottleneck inhibiting their development and practical applications is the scarcity of suitable anode materials. This manuscript addresses this challenge head-on by introducing a novel class of anode material based on a V-based MXene, in conjunction with an ammonium acetate electrolyte. The synergy between these components facilitates a pseudocapacitive mechanism, leading to unprecedented capacity and extended cycling performance.

The authors employ a combination of first-principles calculations and pertinent experimental investigations to elucidate their findings and draw reasonable conclusions. Overall, the manuscript is well-written and accessible, and the subject matter is of significant interest, making it a strong candidate for publication in Nature Communications, where it can garner broader attention from the scientific community. However, in my opinion, certain aspects of the results and analysis warrant more extensive elaboration, and the discussion appears somewhat preliminary. This limits the broader applicability of the conclusions and fails to adequately address key aspects of the study. Therefore, I recommend a thorough revision in this direction, following the guidelines outlined below, as well as addressing some minor issues to enhance the overall clarity of the work.

(1) How might the presence of F and potentially hydroxide in the experimentally realized V₂CT_x material, as revealed, for instance, through XPS, impact the work function and, consequently, the rationale behind the anticipated large working potential windows? It would be highly beneficial to include computational models for F- and, ideally, OH-containing materials, including screening for optimal positions of F and OH species at various meaningful concentrations within sufficiently large supercells. Additionally, conducting an analysis of the electronic electrostatic potential average on the most stable configurations would further enhance the understanding of these effects.

(2) The sentence, 'As a consequence, the absorption of [NH₄⁺] ions on the surface of V₂CO₂ was feasible,' could be misleading. It's crucial to offer a more comprehensive explanation of this feasibility, especially in the context of potential competing adsorption modes and the precise definition of the

adsorption energy. It is particularly important to clarify these aspects as the methods section lacks details regarding the adsorption energy calculation. It appears that the models assume a gas-phase adsorption model, but it remains unclear which non-adsorbed systems are used for computing adsorption energies, such as whether it involves the bare surface and gas-phase NaH_4^+ /acetate molecules. Additionally, the paper should address how it deals with the challenges of modeling a charged system under periodic boundary conditions, particularly with regard to the positive charge on ammonia. It might also be pertinent to consider the inclusion of solvent effects, even if only through the incorporation of an implicit solvent, such as using VASPsol, to enhance the realism of the simulations.

(3) Furthermore, it's important to note that the pseudocapacity mechanism is only activated upon saturation of ammonia adsorption. This implies a qualitative difference between the material model used for computations (low adsorption limit) and the actual experimental scenario of high surface coverage. This requires thorough explanation and, ideally, simulations at higher surface coverage. Moreover, the presence of F on the surface is not investigated in the paper. It could be worthwhile to introduce some F atoms on the surface and examine their role in the adsorption process.

(4) How was the configurational space of the adsorbed clusters explored? Did you use a random generation of a multitude of different structures, and if so, how many structures were generated and what protocol was followed for this purpose? Incorporating some form of molecular dynamics annealing could enhance the exploration process. Additionally, it would be beneficial to address the role of the coordination of nearby ammonia species within the clusters (i.e., high coverage models resembling saturation limit as already mentioned in previous point 3).

(5) When discussing the computed DOS results, the paper touches upon very minor differences that fall within the accuracy of the computational method. However, it is crucial to provide indications of the convergence of the computational setup. For instance, how are these values and trends affected by a larger k-point mesh, which is typically necessary for achieving accurate DOS results? Have the authors considered alternative functionals such as meta-GGA or hybrid functionals, which might yield more reliable results? Similarly, the discussed shift in Bader charges from +1.8 to +1.7 appears quite small. To establish the reliability of these results, it's key to confirm if the results have converged against the resolution grid: it's unclear if a fine FFT grid was employed to accurately reproduce the correct charge distribution. It's advisable to perform a few calculations, systematically increasing $\text{NG}(X,Y,Z)_F$ until the total charge is precisely correct.

(6) It's not entirely clear why V was chosen for the anode material, especially given its presence in cathode materials, as stated. And the manuscript lacks a clear explanation of why this material, used as an "example", ultimately demonstrates impressive performance. Moreover, to improve clarity, it would be beneficial to define "T" as O, OH, and/or F earlier in the text for better comprehension. In this regard, in the DFT calculations, it's important to note that only $T = \text{O}$ is actually computed. Therefore, the generalization to OH and F shown in Figure 1a can be misleading. To enhance clarity, it should be clarified early on in the text that only one system is being considered.

(7) Similarly, why was acetate chosen as the electrolyte anion component? It would be helpful to clarify why alternatives, such as formate, were not considered. What makes acetate unique, and why is it desirable in the context of this study (e.g., cost-effective, sustainable, or other specific advantages)? Overall, I find that the authors haven't sufficiently elucidated what sets acetate apart from other possible anions. While the effect is clearly observed, the electronic structure calculations do not provide much insight into why acetate is distinctive beyond showcasing potential adsorption configurations. It is important to consider that other anions might enable similar adsorption configurations in principle. Therefore, it's crucial to explore and discuss what specifically differentiates acetate from other alternatives. Additionally, some speculative discussion on alternative chemistries that could achieve similar effects as acetate would enhance the completeness of the study.

(8) Why do the figures 1b and 1c display minimums in the average electrostatic potential at the positions of Mo, S, V, and T, while showing a maximum at the position of C? What is the underlying reason for this intriguing electrostatic pattern?

(9) The supplementary figures referenced in the manuscript as 15, 16, 17, 18, and 19 appear to be misaligned by two units when compared to the actual figures presented in the supplementary materials. For example, figure 15 corresponds to what is actually figure 13, and this discrepancy persists throughout the supplementary materials.

Point to point response

Reviewer #1: In this manuscript, the authors reported a novel, pseudocapacitive typed energy storage behavior with unique acetate ion enhancement effect in V₂CT_x MXene. It is impressive that its specific capacity surpasses all of the capacitive-typed electrodes in ammonium-ion battery up to date. Furthermore, **the insight on acetate ion enhancement effect shows general interests on electrolyte design for aqueous ion battery.** Therefore, **I consider it is a nice paper and recommend its publication after some minor revision:**

Response: Thank you so much for your encouragement and positive assessment of our work. Really hope that the revision could address the concerns and make it suitable for publication.

Comment 1: Is the conclusion applicable to other MXenes, such as the common Ti₃C₂T_x?

Response: Thanks for your valuable suggestion. We have tested the specific capacity of the other two MXenes in 0.5 M NH₄Ac electrolyte, Ti₃C₂T_x and V₄C₃T_x, respectively. The specific capacity of Ti₃C₂T_x (30.5 mAh g⁻¹ at 1 A g⁻¹) is much inferior to that of V₄C₃T_x (66.5 mAh g⁻¹ at 1 A g⁻¹), which is because Ti has a poorer valence state than V (Fig R1). The specific capacity of V₄C₃T_x is slightly inferior to that of V₂CT_x, which may be due to the lower V content in V₄C₃T_x compared to V₂CT_x. This indicates that our conclusion is effective for V-based MXenes.

Fig R1. Cycling performance of $\text{Ti}_3\text{C}_2\text{T}_x$ and $\text{V}_4\text{C}_3\text{T}_x$ electrode in 0.5 M NH_4Ac at 1 A g^{-1} .

Comment 2: It is required to give more information about BET results. Also, the equipment information and the preparation of the BET samples should be provided;

Response: Thanks for your reminder. Sorry for our carelessness. We have added the preparation of the BET samples in the revised manuscript (**see red words on Page 24**).

Comment 3: In Supplementary Fig 4b, V exhibits three valence states, why only two valence states exist in ex-situ XPS (Supplementary Fig 13)?

Response: Thanks for your insightful comment. In the initial V_2CT_x , V has three valence states, which we have labeled as V(II), V(III), and V(IV) based on previous reports (*ACS Nano*, 16, 2022, 2, 2711–2720). After the first cycle of charging and discharging, the low valence state of V(II) is completely oxidized during the charging process, and at this time, there are only two valence states of V in the material: V(III) and V(IV) (point A on Supplementary Fig 19a). In the subsequent charging and discharging process, the material cannot be reduced to its initial chemical state, so V only exhibits two chemical valence states.

Comment 4: Please supplement the material characterization after cycling to further demonstrate its stability.

Response: Thanks for your very profound suggestion. The morphology of the $\text{m-V}_2\text{CT}_x$ after 100 cycles is shown in the following SEM picture (Fig R2, and **Supplementary Fig. 9**). After cycles, the $\text{m-V}_2\text{CT}_x$ still maintained a layered structure, indicating its excellent stability.

Fig R2. SEM morphology of m-V₂CT_x after 100 cycles in 0.5 M NH₄Ac at 1 A g⁻¹.

Comment 5: In Fig 3a, has the author attempted to match a full cell with positive electrode materials other than NMO?

Response: Thanks for your comment. We also attempted to use Prussian white (CuHCF) as the positive electrode. However, CuHCF has a low specific capacity, especially at a large current density (58.8 mAh g⁻¹ at 0.3 A g⁻¹, *ACS Appl. Energy Mater.* 2019, 2, 6984–6989). So, its compatibility with V₂CT_x is poor. Taking these factors into consideration, we chose NMO, which has a higher specific capacity with a relatively high electrochemical window, as the positive electrode material for the construction of the full cell.

Comment 6: Please give the capacitive-controlled contribution in other electrolytes and compare them with that in NH₄Ac.

Response: Thanks for your insightful comment. We have tested the CV curves of d-V₂CT_x in the 0.25 M (NH₄)₂SO₄ electrolyte and calculated the capacitive storage contributions in the energy storage process. From the CV curves in Figure R3a, we did not detect the broad redox peaks, which indicated that a similar pseudocapacity did not

occur in this electrolyte. The capacitive-controlled process contributes 43%, 45%, 54%, 69% to the total capacity of d-V₂CT_x at 1, 2, 5, and 10 mV s⁻¹ (Fig R3b, **Supplementary Fig. 16**). Based on these data, the surface-controlled capacity was evaluated to be 46.5 mAh g⁻¹ in 0.5 M NH₄Ac electrolyte (Fig R4, **Supplementary Fig. 17**), which was higher than the 35.3 mAh g⁻¹ in 0.25 M (NH₄)₂SO₄ electrolyte. All of the results are consistent with our conclusion and confirm the acetate ion enhancement effect in NH₄Ac electrolyte. This discussion has been added to the manuscript (**red words on page 13**).

Fig R3. (a) CV curves at different scan rates and (b) the capacitive-controlled contribution of d-V₂CT_x in 0.25 M (NH₄)₂SO₄ electrolyte.

Fig R4. Contributions of d-V₂CT_x in 0.5 M NH₄Ac and 0.25 M (NH₄)₂SO₄ electrolytes, respectively.

Comment 7: Please provide the specific capacity of MXene in high concentration NH₄Ac electrolyte.

Response: Thanks for your suggestion. We have tested the specific capacity of d-V₂CT_x in 20 M NH₄Ac electrolyte (Fig R5, **Supplementary Fig. 13, red words on page 12**). According to our previous CV measurements (Supplementary Fig. 12), it was worth noting that the working potential window (-1 ~ -0.01 V vs. Ag/AgCl) did not apply to this high concentration NH₄Ac electrolyte. A working potential window within -1 ~ -0.1 V vs. Ag/AgCl was used in this test. The specific capacity of d-V₂CT_x in 20 M NH₄Ac electrolyte (88.6 mAh g⁻¹ at 1 A g⁻¹) is much inferior to that in 0.5 M NH₄Ac electrolyte (115.9 mAh g⁻¹ at 1 A g⁻¹). These results indicate that high concentrations of NH₄Ac electrolyte are not conducive to the ammonium-ion storage performance of V₂CT_x.

Fig R5. Cycling performance of d-V₂CT_x electrode in 20 M NH₄Ac at 1 A g⁻¹.

Reviewer #2

Ammonium ion batteries represent an intriguing prospect in the realm of electrochemical energy storage technologies for the future. However, a critical bottleneck inhibiting their development and practical applications is the scarcity of suitable anode materials. This manuscript addresses this challenge head-on by introducing a novel class of anode material based on a V-based MXene, in conjunction with an ammonium acetate electrolyte. The synergy between these components facilitates a pseudocapacitive mechanism, leading to unprecedented capacity and extended cycling performance. The authors employ a combination of first-principles calculations and pertinent experimental investigations to elucidate their findings and draw reasonable conclusions. **Overall, the manuscript is well-written and accessible, and the subject matter is of significant interest, making it a strong candidate for publication in Nature Communications, where it can garner broader attention from the scientific community.**

Response: Thank you so much for your encouragement and positive assessment of our work. In light of the constructive feedback you've offered, we are committed to thoroughly revising our work to ensure that it meets the standards for publication. We are so thankful for the opportunity to refine our work based on your thoughtful comments.

Comment 1: How might the presence of F and potentially hydroxide in the experimentally realized V₂CT_x material, as revealed, for instance, through XPS, impact the work function and, consequently, the rationale behind the anticipated large working potential windows? It would be highly beneficial to include computational models for F- and, ideally, OH-containing materials, including screening for optimal positions of F and OH species at various meaningful concentrations within sufficiently large supercells. Additionally, conducting an analysis of the electronic electrostatic potential average on the most stable configurations would further enhance the understanding of these effects.

Response: Thanks for your suggestion. It is true, and also widely accepted that the surface terminations of MXene materials are generally composed of -F, -O, and -OH,

among which -O is proposed to play a significant role in the redox reaction of MXene electrode. To raise the content of -O termination, an ambient temperature at 90 °C was maintained in the etching procedure, which was proposed to favor the termination transformation from -F/-OH to -O. Therefore, V_2CO_2 was adopted as the major project in DFT simulation. In the first figure of DFT modeling, we agree with you that -F and -OH terminations should be considered, for a comprehensive understanding of the effect of surface conditions on work function. Firstly, the optimal positions of all surface terminations were revealed by comparing the energy of Hollow, V-top, and C-top configurations (Fig R6, **Supplementary Fig. 1**, and Table R1). It can be seen that all three models achieved their minimum energy at the Hollow site. Accordingly, the work functions of three optimal models were calculated, to be 6.62 eV, 5.40 eV, and 1.85 eV for V_2CO_2 , V_2CF_2 , and $V_2C(OH)_2$, respectively. Notably, $V_2C(OH)_2$ exhibits no capability adsorbing NH_4^+ which shows positive adsorption energy while the values in the cases of V_2CO_2 , and V_2CF_2 are computed to be -2.74 eV and -2.09 eV, respectively. As a consequence, the existence of -F with a lower work function would bring a positive shift to the electrochemical window, which has been successfully proved in our HF-etched V_2CT_x under room temperature (-0.8 ~ -0.01 V, Fig R7). We have now replaced **Fig 1 on page 4** in the manuscript, showing the electronic electrostatic potential average of V_2CO_2 and V_2CF_2 , and the “effective” work function of V_2CT_x (T=-O, -F) which dominates the electrochemical window should be located in the range of 5.40 ~ 6.62 eV.

Fig R6. Possible positions of surface terminations on V_2CT_x : Hollow, V-top, and C-top together with the computation results of the electrostatic potential of V_2CT_x (T=-O, -F, -OH) at Hollow site (energetically preferred, ground state).

Table R1. The total energy, work function, and adsorption energy calculation results of V_2CT_x .

	V_2CO_2	V_2CF_2	$V_2C(OH)_2$
Total Energy-Hollow	-44.40 eV	-39.17 eV	-51.22 eV
Total Energy-C top	-43.67 eV	-38.73 eV	-50.95 eV
Total Energy-V top	-41.74 eV	-38.26 eV	-49.45 eV
Work function	6.62 eV	5.40 eV	2.28 eV
Adsorption energy of NH_4^+	-3.63 eV	-2.54 eV	No adsorption

Fig R7. The electrochemical window (-0.8 ~ -0.01 V) of V_2CT_x etched using HF solution.

Comment 2: The sentence, 'As a consequence, the absorption of $[NH_4^+]$ ions on the surface of V_2CO_2 was feasible,' could be misleading. It's crucial to offer a more comprehensive explanation of this feasibility, especially in the context of potential competing adsorption modes and the precise definition of the adsorption energy. It is particularly important to clarify these aspects as the methods section lacks details regarding the adsorption energy calculation. It appears that the models assume a gas-phase adsorption model, but it remains unclear which non-adsorbed systems are used for computing adsorption energies, such as whether it involves the bare surface and gas-phase NH_4^+ /acetate molecules. Additionally, the paper should address how it deals with the challenges of modeling a charged system under periodic boundary conditions, particularly with regard to the positive charge on ammonia. It might also be pertinent to consider the inclusion of solvent effects, even if only through the incorporation of an implicit solvent, such as using VASPsol, to enhance the realism of the simulations.

Response: We would like to express our sincere appreciation to the reviewer for providing such a constructive suggestion for improving simulation works. In experiments, the V_2CT_x electrode was soaked in 0.5 M NH_4Ac electrolyte, which was hydrolyzed to NH_4^+ and Ac^- in solution. Then, the adsorption behavior of NH_4^+ , for

example, was simulated according to:

We agree with you that the state of the ammonium ion should be carefully discussed. In our previous version, the total energy of NH_4^+ is obtained by computing the energy of an NH_4 molecule in a large cell ($15 \text{ \AA} \times 15 \text{ \AA} \times 15 \text{ \AA}$), which gives an energy evaluating the thermal effect relating to the adsorption of NH_4 molecule from the vacuum on V_2CT_x (gas-phase adsorption). In the revised manuscript, we have adopted the VASPsol method to compute the total energies, as recommended by you, with results given in Table R2. The negative values demonstrate that the above reaction is energetically feasible, showing a negative reaction enthalpy for the adsorption process in solution. The simulation methodology has been improved accordingly, as highlighted in the manuscript. The charge on NH_4^+ is indeed neglected in a standard VASP simulation. To our knowledge, the surface of MXene electrode is generally negative (characterized by zeta potential experiment) which is not considered in DFT either. Therefore, there exists an extra coulomb interaction between the negative V_2CT_x and positive NH_4^+ in addition to the computed electronic interaction, which actually favors the adsorption process. As a result, the adsorption energy value gives a low boundary of the adsorption ability.

Table R2. The total energy calculation results using VASPsol.

Models	$E_{\text{total}}^{\text{Vac}}$ (eV)	$E_{\text{total}}^{\text{Sol}}$ (eV)	Solvation energy (eV)
H_2O	-14.22	-14.53	-0.31
NH_4^+	-20.91	-21.45	-0.54
HAc	-46.72	-47.04	-0.32
V_2CO_2 (001)	-44.22	-44.38	-0.16

Comment 3: Furthermore, it's important to note that the pseudocapacity mechanism is only activated upon saturation of ammonia adsorption. This implies a qualitative difference between the material model used for computations (low adsorption limit)

and the actual experimental scenario of high surface coverage. This requires thorough explanation and, ideally, simulations at higher surface coverage. Moreover, the presence of F on the surface is not investigated in the paper. It could be worthwhile to introduce some F atoms on the surface and examine their role in the adsorption process.

Response: Thank you for the profound suggestion. The adsorption limit in our DFT simulation is indeed low, with one NH_4 molecule placed on the surface of a $3 \times 3 \times 1$ supercell of V_2CO_2 . As recommended by you, we have then conducted a molecular dynamics simulation for a thorough understanding of the adsorption behaviors. A large supercell with the size of $12 \times 12 \times 1$ was built in a $35 \times 35 \times 35 \text{ \AA}^3$ box, with a chemical formula of $\text{V}_{288}\text{C}_{144}\text{T}_{288}$ ($\text{T} = \text{-F, -O, -OH}$). Thereafter, the Metropolis Monte Carlo method was employed to introduce 386 NH_4^+ ions or 161 NH_4Ac group. The classical molecular dynamics (MD) simulations were carried out using the LAMMPS code in this work, using the Universal force field and CVFF force field. All the simulations were carried out with a time step of 1 fs. All the systems were relaxed for 1000 ps under isothermal-isobaric ensemble. The temperature was controlled at 300 K using the Nosé-Hoover thermostat with a temperature damping parameter of 0.1 ps and the pressure was also controlled using the Nosé-Hoover barostat with a pressure damping parameter of 1 ps. It can be apparently seen in Fig. R8(**Supplementary Fig. 23**) that an ordered layer of NH_4^+ ions is adsorbed on the surface of V_2CO_2 and V_2CF_2 , which remains coarse in the case of $\text{V}_2\text{C}(\text{OH})_2$, consistent with our previous formation energy results given by DFT. Thereafter, the radial distribution functions (RDFs) were statistically calculated on the last 500ps of relaxation. It can be seen in Fig. R9(**Supplementary Fig. 24**) that a typical bond length of F-H in $\text{V}_2\text{CF}_2 \cdot \text{NH}_4$ is 1.74 \AA , while that of O-H in $\text{V}_2\text{CO}_2 \cdot \text{NH}_4$ is 1.63 \AA , implying a stronger binding energy between V_2CO_2 and NH_4^+ . Furthermore, the numbers of O-H/F-H with a bond length less than 2 \AA (the length of hydrogen bonds in water) in both models were counted. In the case of $\text{V}_2\text{CO}_2 \cdot \text{NH}_4$, the number of O-H bonds achieved stable around 280, which was similar to the number of O atoms, indicating that almost each O atom has bonded with H of NH_4^+ ion, forming a saturated adsorption model close to $\text{V}_2\text{CO}_2 \cdot 2\text{NH}_4$. On the other hand, the number of F-H bonds in the case of $\text{V}_2\text{CF}_2 \cdot \text{NH}_4$ reached stable around 150, indicating that the saturation

adsorption limitation comes when half-F terminations were covered. It can be concluded that the presence of -F termination is unfavorable for the adsorption of NH_4^+ on the surface of V_2CT_x , with smaller adsorption energy and a lower coverage rate of saturation adsorption. Based on the results of the MD simulation, we have revised **Fig 5 and related discussions** in the manuscript. **Please refer to page 17.**

Fig R8. MD simulation of the adsorption behavior of NH_4^+ on the surface of V_2CT_x (T=-F, -O, -OH).

Fig R9. Bond length statistics from MD simulation with the adsorption of NH_4^+ on the surface of V_2CT_x ($T=-\text{F}, -\text{O}, -\text{OH}$).

Comment 4: How was the configurational space of the adsorbed clusters explored? Did you use a random generation of a multitude of different structures, and if so, how many structures were generated and what protocol was followed for this purpose? Incorporating some form of molecular dynamics annealing could enhance the exploration process. Additionally, it would be beneficial to address the role of the coordination of nearby ammonia species within the clusters (i.e., high coverage models resembling saturation limit as already mentioned in previous point 3).

Response: Thank you for the suggestion. The adsorption configurations in this work were investigated using the DFT method. We agree with you that the incorporation of some MD simulations could enhance the exploration process. Therefore, two methods using DFT and MD have been performed to find out the possible configurations. In Materials Studio, an optimized $[\text{NH}_4]$ molecule was randomly placed above the V_2CT_x supercell, and assigned to be a motion set that is free to move and rotate as a group. During the structure optimization process, the $[\text{NH}_4]$ molecule would always find its energetically optimal position regardless of its initial positions, corresponding to the ground state of the investigated model, the Hollow configuration displayed in Fig 5c. To obtain other “metastable” adsorption sites, the x - y fraction coordinates of central N

were then fixed, and the four hydrogen atoms together with the N-z coordinate were relaxed, achieving the total energies of “O mid” and “O top” configurations. In LAMMPS, a number of NH_4^+ ions were placed into the vacuum space of the V_2CT_x supercell using the Metropolis Monte Carlo method. After a total relaxation, NH_4^+ ions close to the surface of the V_2CT_x layer were collected, and the bonding feature was revealed. Similar results can be obtained from MD simulation with the DFT calculation, showing three typical bonding configurations as displayed in Fig. R10(Supplementary Fig. 20).

Fig R10. Typical adsorption configurations of $\text{V}_2\text{CO}_2 \cdot [\text{NH}_4^+]$ and $\text{V}_2\text{CO}_2 \cdot [\text{NH}_4^+(\text{HAc})_x]$ models after a full relaxation of MD simulation by LAMMPS.

Comment 5: When discussing the computed DOS results, the paper touches upon very minor differences that fall within the accuracy of the computational method. However, it is crucial to provide indications of the convergence of the computational setup. For instance, how are these values and trends affected by a larger k-point mesh, which is typically necessary for achieving accurate DOS results? Have the authors considered alternative functionals such as meta-GGA or hybrid functionals, which might yield more reliable results? Similarly, the discussed shift in Bader charges from +1.8 to +1.7 appears quite small. To establish the reliability of these results, it's key to confirm if the results have converged against the resolution grid: it's unclear if a fine FFT grid was employed to accurately reproduce the correct charge distribution. It's advisable to perform a few calculations, systematically increasing $NG(X,Y,Z)F$ until the total charge is precisely correct.

Response: Thank you for the suggestion. In the revised manuscript, we have increased the computation precision according to the reviewer's suggestion, with the cutoff energy set to be 800 eV and total k -points set to be 120 (equivalent to a k -mesh of $15 \times 15 \times 5$ in a $3 \times 3 \times 1$ supercell), with the DOS curves displayed in Figure 5f. Minor differences can be found in the PDOS curves of the V element between V_2CO_2 and $V_2CO_2[NH_4^+(HAc)_3]$. Two possible factors are causing this minor difference: firstly, only the PDOS of the V element, instead of the total DOS, is given in the figure, while the bonding of V (to C and O) is similar in these two cases; secondly, a large smearing width (0.1 eV) was once used before. In the present manuscript, we increased the NDOS values using a small smearing width (0.05eV) and added a shadow area showing the total DOS which changes apparently with the presence of $[NH_4^+(HAc)_3]$ on the surface. Generally, the hybrid functionals can be more accurate in evaluating the bandgap of nonmetallic systems. In this work, the investigated V_2CO_2 appears to be metal-like, showing no bandgap at Fermi level. Therefore, we adopted the pure GGA method while computing DOS curves.

We agree with you that the Bader charge of V in V_2CO_2 and $V_2CO_2[NH_4^+(HAc)_3]$ is close, +1.8 versus +1.7. It might be improper to adopt only one effective digit in this case. Therefore, we increased the effective digits to two numbers after the dot, and then

the valence state of the V element changed from +1.78 to +1.69 with the adsorption of $\text{NH}_4^+(\text{HAc})_3$ group on the surface (red words on page 18). Notably, the valence change of -0.09 (11.5%) is obtained with one adsorption group. According to the molecular dynamics analysis, the value of the experiment should be much larger.

Comment 6: It's not entirely clear why V was chosen for the anode material, especially given its presence in cathode materials, as stated. And the manuscript lacks a clear explanation of why this material, used as an “example”, ultimately demonstrates impressive performance. Moreover, to improve clarity, it would be beneficial to define “T” as O, OH, and/or F earlier in the text for better comprehension. In this regard, in the DFT calculations, it's important to note that only T = O is actually computed. Therefore, the generalization to OH and F shown in Figure 1a can be misleading. To enhance clarity, it should be clarified early on in the text that only one system is being considered.

Response: Thank you for the suggestion. We apologize for the inaccuracies in our expression. In ammonium-ion batteries (AIBs), vanadium-based oxides are a common type of material with relatively high specific capacity and rate capability owing to the rich valence state of the V element. Currently, reported vanadium-based cathode materials include V_2O_5 (*Chem*, 2019, 5, 1537–1551), and $\text{NH}_4\text{V}_4\text{O}_{10}$ (*Nano Energy*, 2020, 68, 104369). Although they are all used as cathode in the reported studies, their working potentials ($-0.2 \sim 0.8 \text{ V vs Ag/AgCl}$ for V_2O_5 , and $0 \sim 1 \text{ V vs Ag/AgCl}$ for $\text{NH}_4\text{V}_4\text{O}_{10}$) are generally lower than other cathode materials such as CuHCF ($0.5 \sim 1.5 \text{ V vs Ag/AgCl}$, *ACS Appl. Energy Mater.* 2019, 2, 6984–6989). The working potential of V-based compounds is closer to that of anode materials, such as MoO_3 ($-0.45 \sim 0.85 \text{ V vs Ag/AgCl}$, *Adv. Mater.* 2020, 32, e1907802). Based on these, we aim to design a V-based material for the anode of AIBs with a lower operating voltage while simultaneously maintaining a high specific capacity. Therefore, we selected V_2CT_x MXene as the research sample, which is widely used as an anode material in lithium-ion batteries and sodium-ion batteries. We have made revisions to the explanation in the manuscript, please refer to the red word on page 4.

Regarding the detailed description of terminations, we have addressed it in response to comment 1. The corrections have been also made to **Fig 1**. We hope that the revised version can address your concerns.

Comment 7: Similarly, why was acetate chosen as the electrolyte anion component? It would be helpful to clarify why alternatives, such as formate, were not considered. What makes acetate unique, and why is it desirable in the context of this study (e.g., cost-effective, sustainable, or other specific advantages)? Overall, I find that the authors haven't sufficiently elucidated what sets acetate apart from other possible anions. While the effect is clearly observed, the electronic structure calculations do not provide much insight into why acetate is distinctive beyond showcasing potential adsorption configurations. It is important to consider that other anions might enable similar adsorption configurations in principle. Therefore, it's crucial to explore and discuss what specifically differentiates acetate from other alternatives. Additionally, some speculative discussion on alternative chemistries that could achieve similar effects as acetate would enhance the completeness of the study.

Response: Thank you for the suggestion. That's a good question that has confused us for a long time. It is interesting to find in experiments that only in acetate does the V_2CT_x electrode exhibit excellent performance. However, in DFT simulation, the adsorption of $[NH_4^+(HMc)]$ is also feasible, with the formation energy calculated to be -2.77 eV. Therefore, we believe that the critical factor making acetate distinctive is not from simulation (adsorption energy or electronic structure), but from experiment, for example, pH value as discussed in our manuscript: among the all investigated ammonium ion electrolytes in this work, the pH value of NH_4Ac is 6.5, closest to neutral. As a consequence, NH_4Ac can be hydrolyzed to NH_4^+ and Ac^- in solution which take part in the reaction. In the case of NH_4Mc (0.5 M), the pH value is around 6.0, demonstrating that the electrolyte is rich in H^+ , Mc^- , and $NH_3 \cdot H_2O$.

Comment 8: Why do the figures 1b and 1c display minimums in the average electrostatic potential at the positions of Mo, S, V, and T, while showing a maximum at the position of C? What is the underlying reason for this intriguing electrostatic pattern?

Response: Thank you for the suggestion. The average electrostatic potential represents the energy change while moving a free electron from the sample (bonding) to the vacuum along the projection direction, then the difference between the vacuum and Fermi level represents the work function. Interestingly, few discussions about the inner part can be found in literature. To our best knowledge, the potential curve generally achieves its minimum values at ionic sites, which are related to the element species and valence state: for example, -14.3 eV at S and -17.3 eV at Mo in MoS₂. Then, the combination of three deep valleys brings the potential curve to two peaks between Mo and S atoms (↓↗↘↗↘↑). In the case of V₂CO₂, the curve achieved -17.4 eV at V, -8.7 eV at C, and -7.4 eV at O. Then, the “valleys” combination follows “shallow-deep-shallow-deep-shallow”, leaving only one peak at C site (↓↘↗↘↗↑). It is worthwhile pointing out that the electrostatic potential pattern is also related to the projection direction: a regular peak between V and C elements is possibly observed along the V-C bond direction by adjusting the projection direction.

Comment 9: The supplementary figures referenced in the manuscript as 15, 16, 17, 18, and 19 appear to be misaligned by two units when compared to the actual figures presented in the supplementary materials. For example, figure 15 corresponds to what is actually figure 13, and this discrepancy persists throughout the supplementary materials.

Response: Thanks for your careful review. Sorry for our carelessness. We have checked carefully and corrected the corresponding references in the revised manuscript (please see **Pages 15~16 and supplementary files**).

REVIEWERS' COMMENTS

Reviewer #1 (Remarks to the Author):

All comments have been well revised and now it can be formally accepted.

Reviewer #2 (Remarks to the Author):

The revised manuscript is commendable, with esuccess in addressing previous concerns. In my opinion, the manuscript has reached a publishable standard for Nature Communications. However, I would like to highlight a specific area of improvement in the Computational Methods section to enhance consistency and reproducibility of results.

Specifically, additional details pertaining to the employed force fields are essential. It is recommended to include appropriate references for the specifically utilized UFF and CVFF, ensuring transparency in the methodology. Moreover, for the benefit of readers and researchers aiming to replicate or build upon your work, consider supplementing the manuscript with a table or equivalent supplementary information containing all pertinent force field parameters considered in the simulations. This inclusion will contribute to the robustness and clarity of the research methodology, aligning with the journal's standards.

Point to point response

Reviewer 1: All comments have been well revised and now it can be formally accepted.

Response: Thank you for your diligence in the comments. Your attention to detail is greatly appreciated, and we look forward to moving forward with confidence in the finalized comments. Once again, thank you for your efforts in ensuring the quality of the comments.

Reviewer 2:

The revised manuscript is commendable, with esuccess in addressing previous concerns. In my opinion, **the manuscript has reached a publishable standard for Nature Communications**. However, I would like to highlight a specific area of improvement in the Computational Methods section to enhance consistency and reproducibility of results.

Specifically, additional details pertaining to the employed force fields are essential. It is recommended to include appropriate references for the specifically utilized UFF and CVFF, ensuring transparency in the methodology. Moreover, for the benefit of readers and researchers aiming to replicate or build upon your work, consider supplementing the manuscript with a table or equivalent supplementary information containing all pertinent force field parameters considered in the simulations. This inclusion will contribute to the robustness and clarity of the research methodology, aligning with the journal's standards.

Response: Thank you for your positive feedback. We are delighted that you consider it to meet the standards for publication in Nature Communications. Additionally, we appreciate your identification of potential improvement areas in the Computational Methods section.

We have incorporated relevant references for UFF and CVFF in the method, please see **red words in page 25**. Furthermore, we have added a table in the supplementary information containing all pertinent force field parameter. If there are other aspects that require attention or further improvement, please feel free to let us know. Thank you once again for your valuable suggestions.

Supplementary Table 1 force field parameters considered in the MD simulations

Pair Coefficient	D_0 (kcal/mol)	R_0 (Å)
V, in MXene	0.0266000000	3.1440000000

C, in MXene	0.1050000000	3.8510000000
O, in MXene	0.2280000124	2.8597848722
C, in methyl group	0.0389999952	3.8754094636
C, in charged carboxylate group	0.1479999981	3.6170487995
H, bonded to carbon	0.0000000000	0.0000000000
H, bonded to nitrogen	0.0000000000	0.0000000000
O, in charged carboxylate group	0.2280000124	2.8597848722
N, with 4 substituents	0.1669999743	3.5012320066
F, in MXene	0.0687685101	3.0808078941
Bond Coefficient	k_1 (kcal/mol Å²)	r_0 (Å)
V-C	560.0000	2.4100
V-O	560.0000	2.1200
C-C	283.0924	1.5200
C-H	340.6175	1.1050
C-O	540.0000	1.2500
H-N	457.4592	1.0260
V-F	560.0000	2.0600
Angle Coefficient	k_2 (kcal/mol rad²)	θ_0 (deg)
C-V-C	30.0000	95.5200
C-V-O	30.0000	87.2900
O-V-O	70.0000	109.5000
V-C-V	30.0000	109.4700
V-O-V	60.0000	109.5000
C-C-H	45.0000	109.5000
H-C-H	39.5000	106.4000
C-C-O	68.0000	120.0000
O-C-O	145.0000	123.0000
H-N-H	36.0000	105.5000
C-V-F	30.0000	87.2900
V-F-V	60.0000	109.4700